# Whole genome sequencing of skull-base chordoma reveals genomic alterations associated with recurrence and chordoma-specific survival

Jiwei Bai[1,2,3,9], Jianxin Shi[4,9], Chuzhong Li [1,2,3,5,9], Shuai Wang[1,9], Tongwu Zhang [4], Xing Hua[4], Bin Zhu [4], Hela Koka[4], Ho-Hsiang Wu [4], Lei Song[4,6], Difei Wang[4,6], Mingyi Wang [4,6], Weiyin Zhou[4,6], Bari J. Ballew[4,6], Bin Zhu[4,6], Belynda Hicks [4,6], Lisa Mirabello[4], Dilys M. Parry[4], Yixuan Zhai[1,7], Mingxuan Li[1], Jiang Du[1,3,5], Junmei Wang[1,3,5], Shuheng Zhang[1,8], Qian Liu[1], Peng Zhao[2,3], Songbai Gui[2,3], Alisa M. Goldstein [4], Yazhuo Zhang [1,2,3,5✉] & Xiaohong R. Yang[4]

Chordoma is a rare bone tumor with an unknown etiology and high recurrence rate. Here we conduct whole genome sequencing of 80 skull-base chordomas and identify *PBRM1*, a SWI/SNF (SWItch/Sucrose Non-Fermentable) complex subunit gene, as a significantly mutated driver gene. Genomic alterations in *PBRM1* (12.5%) and homozygous deletions of the *CDKN2A/2B* locus are the most prevalent events. The combination of *PBRM1* alterations and the chromosome 22q deletion, which involves another SWI/SNF gene (*SMARCB1*), shows strong associations with poor chordoma-specific survival (Hazard ratio [HR] = 10.55, 95% confidence interval [CI] = 2.81-39.64, p = 0.001) and recurrence-free survival (HR = 4.30, 95% CI = 2.34-7.91, p = $2.77 \times 10^{-6}$). Despite the low mutation rate, extensive somatic copy number alterations frequently occur, most of which are clonal and showed highly concordant profiles between paired primary and recurrence/metastasis samples, indicating their importance in chordoma initiation. In this work, our findings provide important biological and clinical insights into skull-base chordoma.

[1] Beijing Neurosurgical Institute, Capital Medical University, Beijing, China. [2] Department of Neurosurgery, Beijing Tiantan Hospital, Capital Medical University, Beijing, China. [3] China National Clinical Research Center for Neurological Diseases, Beijing, China. [4] Division of Cancer Epidemiology and Genetics, National Cancer Institute, NIH, DHHS, Bethesda, MD, USA. [5] Brain Tumor Center, Beijing Institute for Brain Disorders, Beijing, China. [6] Cancer Genomics Research Laboratory, Leidos Biomedical Research, Frederick National Laboratory for Cancer Research, Frederick, MD, USA. [7] Department of Neurosurgery, The First Affiliated Hospital of Zhengzhou University, Zhengzhou, China. [8] Department of Neurosurgery, Anshan Central Hospital, Anshan, China. [9] These authors contributed equally: Jiwei Bai, Jianxin Shi, Chuzhong Li, Shuai Wang. ✉email: zhangyazhuo@ccmu.edu.cn

Chordoma is a rare bone tumor, which is believed to originate from notochordal remnants[1] and occurs in the axial skeleton of cranial, vertebral, and sacral sites[2]. Based on the United States Surveillance Epidemiology and End Results data, the incidence of chordoma varies by gender and race[3], however, little is known about the etiologic factors that predispose to it. Germline duplication of the *TBXT* gene, which encodes brachyury, a transcription factor that plays an important role in embryonic development, was identified as a major susceptibility mechanism in familial chordoma[4]. A common genetic polymorphism in *TBXT* was subsequently associated with an increased risk for both familial and sporadic chordoma[5,6].

Chordomas are considered slow growing; however, the recurrence rate is high, especially among skull-base chordoma patients, largely due to incomplete tumor resection. The clinical progression of skull-base chordoma is highly variable[7], and there are no validated clinical or molecular prognostic panels available. Treatment for a skull-base tumor usually involves surgery with or without adjuvant radiation therapy (RT). Chemotherapy or other systemic therapies are not effective for treating chordoma. Although several potentially druggable molecular targets have been identified and some are being evaluated in clinical trials[8], treatment options for chordoma patients, particularly those with advanced disease, are still limited. A better understanding of the molecular processes in chordoma is critically needed to develop prognostic prediction tools and to discover druggable targets.

Genomic profiling studies of chordoma are limited. The largest sequencing analysis so far included 104 sacral chordoma patients, but whole-genome sequencing (WGS) was conducted on only 11 tumors[9]. Results from this study suggested that amplifications of *TBXT* (encoding brachyury), homozygous deletion of *CDKN2A*, and mutations in SWI/SNF (SWItch/Sucrose Non-Fermentable) complex genes (*PBRM1, SETD2, ARID1A*) and the PI3K signaling pathway were the most frequent genomic events in sacral chordoma. Limited by the small number of WGS and WES samples analyzed, formal testing for driver gene mutations, structural variants, and mutation signatures was not conducted. It also remains unclear whether skull-base chordoma, which is associated with a much earlier age onset (47.4 years) compared to sacral chordoma (62.7 years)[3], is driven by similar genomic alterations. Furthermore, the clinical relevance of these genomic events is largely unknown.

Here in this chordoma WGS study, we provide a detailed genomic landscape of skull-base chordoma, which revealed potential driver events, mutational signatures, and outcome-related genomic features. Our findings suggest that the combination of SWI/SNF alterations and 22q deletion show a strong association with clinical outcomes, demonstrating the potential of designing a multi-marker panel in prognostic prediction.

## Results

**Patient characteristics**. The primary analysis included 80 patients with skull-base chordoma who were diagnosed and treated at Beijing Tiantan Hospital, China. The detailed clinical characteristics of these patients are shown in Table 1. In brief, the mean age at initial diagnosis of chordoma among these patients was 44.7 years (range: 7–79); 62.5% were males, the majority (80%) had conventional/classical chordoma, and 12 received RT prior to surgery. After an average follow-up period of 50 months, there were 59 recurrences and 17 deaths, all died of chordoma (Table 1).

We performed WGS on 91 surgically resected chordomas from 80 patients (including 11 paired primary and recurrent tumor samples) and their matched germline DNA from whole blood, with the average sequencing depth of 41x for blood and 76x for tumor samples (Supplementary Fig. 1). The average tumor purity estimated from copy number (CN) alterations was 57%.

In a separate analysis, we also analyzed a patient with metastatic chordoma, who was diagnosed with chordoma at 17 years old. During post-surgery RT, her tumor was found to have recurred in the skull base and to have metastasized into the lymph node, thorax, and liver. We conducted WGS on the chordoma recurrence sample, lymph node metastasis (LM), thoracic metastasis (TM), and their matched germline blood samples.

**Germline susceptibility**. *TBXT* is the only chordoma susceptibility gene identified to date. Germline *TBXT* gene duplication was reported in a subset of chordoma families[4] but was rare in sporadic chordoma patients[6]. Using matched germline WGS data, we found germline *TBXT* gene duplication in a single patient (P22, Supplementary Fig. 2), who was 47 years old at diagnosis and had local recurrence. We then evaluated rare exonic variants in known cancer predisposition genes[10] or previously reported chordoma-related genes (including potential germline susceptibility and somatically mutated genes, Supplementary Table 1) and found two variants, each in a single patient, that were classified as pathogenic (*ERCC5*, c.697 C > T, p. Gln233*) or likely pathogenic (LP) (*BLM*, c.3564delC, p. Phe1189fs) (see classification criteria in "Methods"). Both

**Table 1 Characteristics of chordoma patients included in the whole-genome sequencing analysis (n = 80)[a].**

| N (%) | |
|---|---|
| Age (year)[b] | 44.7 (7–79) |
| ≤20 | 10 (12.5) |
| 20–40 | 21 (26.2) |
| 40–50 | 14 (17.5) |
| 50+ | 35 (43.8) |
| Sex | |
| F | 30 (37.5) |
| M | 50 (62.5) |
| Histological type | |
| Classic | 64 (80.0) |
| Chondroid | 14 (17.5) |
| Dedifferentiated | 2 (2.5) |
| Surgery type | |
| Endoscopic endonasal | 78 (97.5) |
| Open craniotomy | 2 (2.5) |
| Gross resection rate | |
| Complete | 16 (20.0) |
| Near complete | 36 (45.0) |
| Subtotal or partial | 28 (35.0) |
| Tumor volume (cm³)[b] | 35.0 (4.2–147.5) |
| Presurgery RT[c] treatment | |
| No | 68 (85.0) |
| Yes | 12 (15.0) |
| Post-surgery RT[c] treatment | |
| No | 38 (47.5) |
| Yes | 42 (52.5) |
| Recurrence status | |
| No | 21 (26.3) |
| Yes | 59 (73.7) |
| RFS[d] (month)[b] | 21 (1–128) |
| Death status | |
| No | 63 (78.8) |
| Yes | 17 (21.2) |
| Survival (month)[b] | 50 (10–157) |

[a]None of these patients received any chemo, immune, or targeted therapies.
[b]Characteristic in its continuous form is expressed as mean (range).
[c]Radiation therapy.
[d]Recurrence-free survival.

patients were heterozygous carriers, and the evaluation of somatic mutation and CN data did not find mutations or losses of the other allele in either primary tumor or recurrence samples in these two patients. In summary, our data did not support a strong germline contribution from known cancer predisposition genes or chordoma genes in this patient cohort.

A common genetic variant in the *TBXT* gene (rs2305089) was previously associated with a sixfold increase in risk of developing chordoma in a European population[5], however, this variant was not significantly associated with chordoma risk in a Chinese study of skull-base chordoma[11]. In our study, the frequency of the variant allele (A) among chordoma patients is 33.7%, which is similar to what was reported in the Chinese study by Wu et al.[11] and is much lower than those reported in European chordoma patients (>80%). The frequency of the variant allele observed among chordoma patients in our study is also similar to those reported in the general population among East Asians, further implicating that this variant is not associated with skull-base chordoma among Chinese.

**Somatic genomic landscape**. Among 80 distinct tumors in the primary analysis, most had low tumor mutational burden (TMB, median = 0.53 mutations/Mb, range = 0.05–7.68, per tumor). TMB was lower compared to most cancer types sequenced in The Cancer Genome Atlas (TCGA) (Supplementary Fig. 3). Numbers of single nucleotide variations (SNVs), insertion and deletions (indels), and structural variants (SVs) in these patients are shown in Fig. 1. Patients who received presurgery RT (n = 12) tended to have higher TMB (median = 0.75 mutations/Mb, range = 0.055–1.56) compared with patients without presurgery treatment (n = 68, median = 0.49 mutations/Mb, range = 0.05–7.68), however, the difference was not statistically significant (p = 0.42). The average number of SVs per tumor was 38. The most prevalent SV events included deletions (13%), deletions with insertions (16%), and chromosomal translocations (13% for inter-chromosome and 10% for intra-chromosome translocations) (Fig. 1).

**Mutational signatures**. SigProfiler (https://github.com/AlexandrovLab)[12] was used to identify single-base substitution (SBS), double-base substitution (DBS), and small indel (ID) mutational signatures. We identified nine de novo SBS mutation patterns in 80 primary tumors that were highly correlated with combinations of existing COSMIC signatures (cosine similarities ranging from 0.84 to 0.99; Supplementary Fig. 2A). No novel SBS patterns were identified in this analysis.

Since the observed de novo signatures largely overlapped with known COSMIC signatures, we therefore focused on the contributions of the known signatures to the mutational landscape of our chordoma cohort. The predominant SBS signature is SBS5 (mean fraction = 67.9%, range = 0–94.9%), which is a common clock-like signature in many cancer types. Other prevalent SBS signatures included SBS8 (9.9%), SBS1 (10.0%), and SBS40 (5.4%) (Fig. 1 and Supplementary Table 2A). SBS1 is associated with the deamination of 5-methylcytosine, reflecting age-related accumulation. SBS8, which was recently associated with nucleotide excision repair deficiency in breast tumors[13], was associated with higher TMB (Spearman correlation, p = 0.017). The average fractions of *APOBEC* signatures (SBS2: 1.1% and SBS13: 1.2%), which were low in this patient cohort (Fig. 1 and Supplementary Table 2A), were associated with higher number of SVs (p = 0.002). Two tumors (P21 and P73), in which TMB was among the highest, showed high contributions of SBS44, which is a signature associated with defective DNA mismatch repair (MSI). One of these tumors (P21) carried nonsynonymous mutations in *MLH1* and *TP53*, showing a

characteristic of high MSI. Interestingly, both tumors had alterations in *SETD2* (one missense mutation predicted to be damaging by PolyPhen and Sift and one SV).

We also identified five de novo indel signature patterns, four of which reflected combinations of COSMIC ID1–6, 8, and 9 (cosine similarities ranging from 0.93 to 1.00; Supplementary Fig. 4B). The remaining one (signature A), which is composed predominantly of 2 bp insertions (mainly AT and TA) at long (≥5) repeats, was not mapped to any COSMIC ID signatures (Fig. 1). This signature was prevalent in chordoma tumors, which was present in 55 of 80 tumors (Supplementary Table 2B). The number of mutations per tumor sample attributed to this signature was not associated with any patient characteristics or genomic features such as age, sex, presurgery or post-surgery RT, TMB, or any SBS/DBS signatures.

When conducting the mutational signature analysis separately for clonal and subclonal mutations classified by PyClone[14], we found that fractions of clonal and subclonal signatures were highly correlated. No significant differences were found between clonal and subclonal analyses (Supplementary Fig. 5).

When comparing the mutational profiles and signatures of chordoma against those of 9450 tumor samples comprised of different cancer types using MutaGene (https://www.ncbi.nlm.nih.gov/research/mutagene/)[15], we found that the mutational profiles of chordoma were most similar to those of kidney cancers (clear cell and papillary cell), chronic lymphocytic leukemia, and bone cancer (Supplementary Table 3).

**Recurrently mutated genes**. When restricting to nonsynonymous mutations (see "Methods" for definition), we found that *PBRM1* (6.25%), *B2M* (3.75%), and *MAP3K4* (3.75%) were the most frequently mutated known cancer driver genes[16] in this patient cohort. Driver gene analysis combining SNVs and indels using dNdScv (v0.1.0, https://github.com/im3sanger/dndscv/releases/tag/0.1.0)[17] identified only one significantly mutated gene, *PBRM1*, at false discovery rate (FDR) q < 0.1 (q = 0.0001). Five different *PBRM1* mutations were found, each in a different patient (Fig. 2). We also found two mutations in *LYST* (each in a single tumor), a gene that encodes a protein regulating intracellular protein trafficking in endosomes and was previously suggested as a potential chordoma driver gene[9]. One of the two *LYST* mutation carriers also harbored a germline *TBXT* duplication. *TP53* mutations, which are common across different cancer types, were only seen in one chordoma patient. Mutations in other known cancer driver genes were detected in 17 additional tumors, each only occurred in a single sample (Fig. 2), indicating substantial heterogeneity in the driver mutation landscape.

For mutations in noncoding regions, we focused on mutations in the promoter region of known cancer driver genes, and we did not detect mutations in any of these genes including *TERT* in our patient cohort.

**Somatic copy number alterations (SCNAs) and structural variants (SVs)**. Using FACETS[18], we identified arm/chromosome-level SCNAs in the majority (77.5%) of primary chordoma samples. Consistent with previous reports[19,20], we observed frequent arm-level SCNA events (Fig. 3a). Among them, 17 events were found to be significant (q < 0.1) by GISTIC (https://software.broadinstitute.org/cancer/cga/gistic) analysis[21], including gains of chromosomes 1q, 7p, and 7q, and deletions of 1p, 3, 4, 9, 10, 13q, 14q, 18, and 22q. Clustering analysis revealed five distinct groups of patients based on SCNA events. Group 1 (n = 16) and Group 2 (n = 25) demonstrated extensive SCNAs, with Group 2 lacking chromosome gains and Group 1 lacking deletions of 4, 9, and 14. Group 3 (n = 13) had scattered SCNAs, most of which are

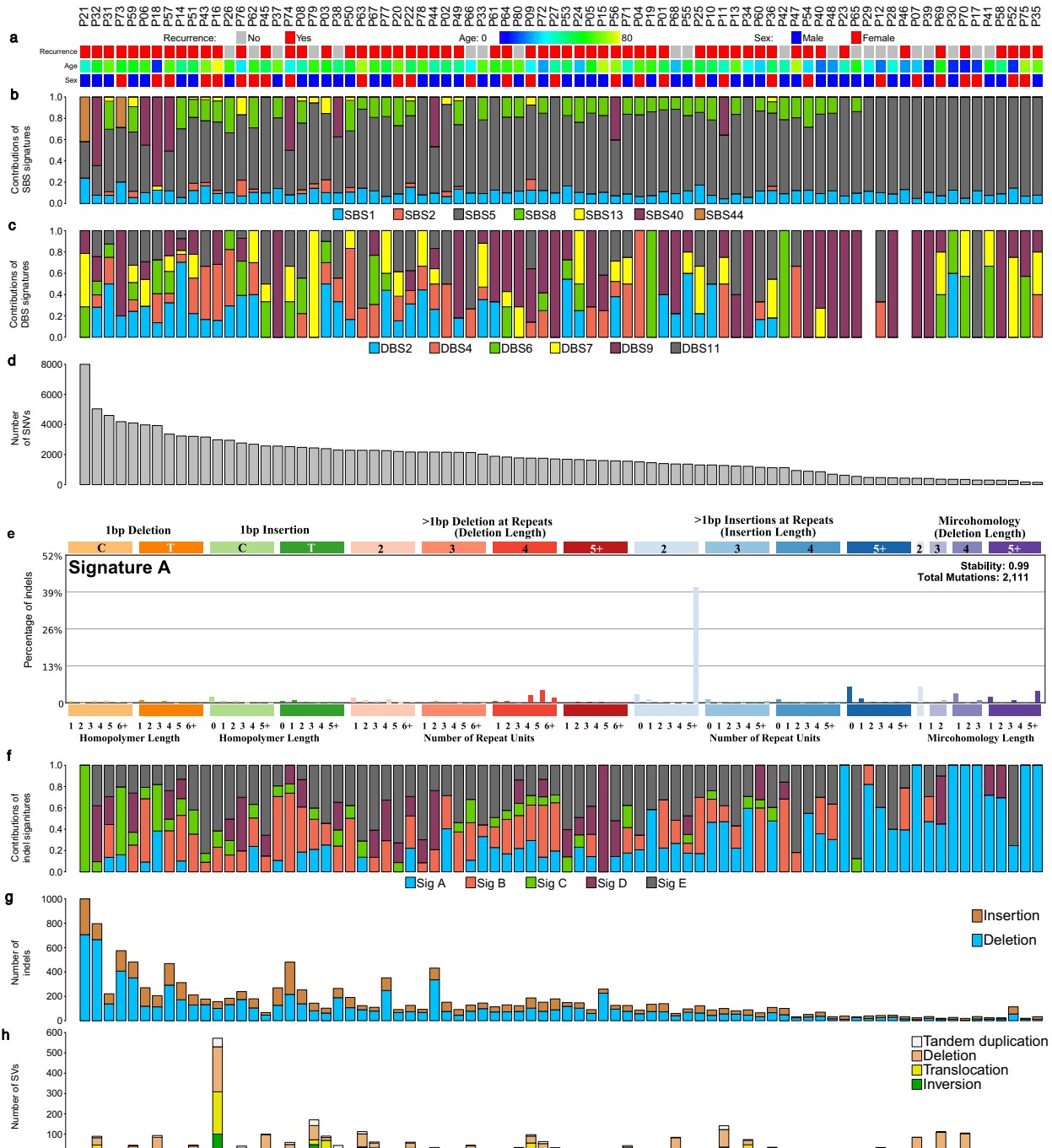

**Fig. 1 Mutational landscape of skull-base chordoma (*n* = 80). a** The recurrence status (no: gray; yes: red), age (from younger to older: blue to yellow), and sex (female: red; male: blue); **b** the proportions of somatic single-base substitutions (SBS) found in each chordoma patient that can be attributed to COSMIC SBS signatures; **c** the proportions of somatic double-base substitutions (DBS) found in each chordoma patient that can be attributed to COSMIC DBS signatures; **d** number of single nucleotide variants (SNVs) identified in each chordoma sample; **e** the de novo signature A, which is not mapped to any of COSMIC indel signatures; **f** the proportions of de novo indels found in each chordoma patient; **g** number of indels found in each chordoma sample; **h** number of structural variants (SVs) found in each chordoma sample. See also Supplementary Figs. 4 and 5 and Tables 2 and 3.

deletions. Group 4 (*n* = 19) had no or few SCNAs. Group 5 (*n* = 7) was characterized by the enrichment for extensive chromosome gains throughout the genome (Fig. 3a), a characteristic consistent with whole-genome doubling (WGD), defined by 50% of autosomal genome having an MCN (the more frequent allele in a given segment) greater than or equal to two (Supplementary

Table 4)[22]. The number of nonsynonymous SNVs and SVs did not vary significantly among the five SCNA groups.

GISTIC analysis of focal SCNA regions identified 6 significant (*q* < 0.1) amplifications and 12 significant deletion regions (Supplementary Fig. 6A). The most significant deletion region was 9p21.3 (*p* = 5.87 × 10$^{-13}$), which contains the known tumor

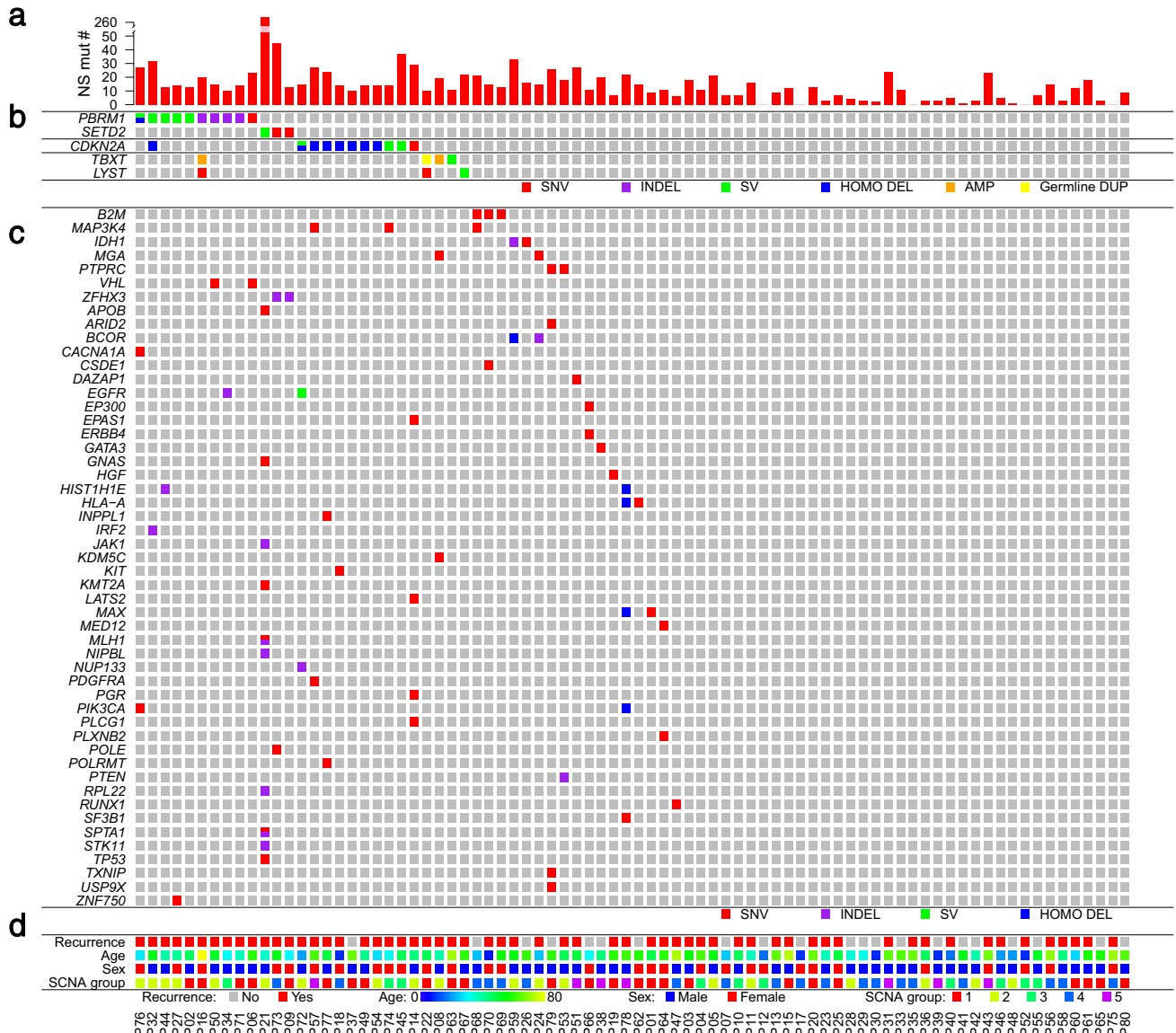

**Fig. 2 Genomic driver landscape of skull-base chordoma. a** The number of nonsynonymous (NS) mutations per tumor. **b** The potential chordoma driver genes. **c** The middle gene panel reports NS mutations in known cancer driver genes that were detected in this study. SNV single nucleotide variant, SV structural variant, HOMO DEL homozygous deletion, AMP amplification, Germline DUP germline duplication. **d** The recurrence status (no: gray; yes: red), age (from younger to older: blue to yellow), sex (female: red; male: blue), and SCNA group (see Fig. 3). See also Supplementary Fig. 2.

suppressor gene *CDKN2A*, while the peak region of the 3p21.1 deletion ($p = 0.062$) contained *PBRM1* and *SETD2*, both are chromatin remodeling genes. In a subset of patients with RNA sequencing (RNA-Seq) data available ($n = 27$), we found that patients with 3p21.1 deletion had decreased expression of *PBRM1* ($p = 0.025$) and *SETD2* ($p = 0.033$) (Supplementary Fig. 6B). In contrast to reported frequent somatic duplications of *TBXT* (>20%) as a driver event in sacral chordoma[9], focal *TBXT* duplications were only seen in two tumors (P08 and P16) in our patient cohort, which did not reach statistical significance.

We used the Meerkat algorithm[23] to identify SVs and used ShatterSeek (v0.4) to detect chromothripsis (clusters of SVs)[24]. Similar to what was previously reported in glioblastoma[23], we found complex genomic rearrangement events involving the *CDKN2A/2B* locus (Fig. 3b). Thirty-one (40%) tumors had arm-level 9p deletion, 9 of which also had focal deletions or complex rearrangements of the 9p21.3 region resulting in the homozygous loss of the *CDKN2A/2B* locus (Fig. 3b).

Twelve high-confidence chromothripsis events were detected in seven tumors (Supplementary Table 5), with two events (in P16 and P25) involving chromosome 3p, including the *PBRM1* gene region (Fig. 3c). We also observed extensive chromothripsis of the chromosome 6q region encompassing the *TBXT* gene in one tumor (P63, Fig. 3d).

**Summary of driver landscape of skull-base chordoma.** To characterize the potential driver events for skull-base chordoma, we combined five patients with *PBRM1* mutations and five patients with SVs involving *PBRM1* (Fig. 4), which were validated by targeted sequencing. Together with *SETD2* alterations in three patients, alterations in these two chromatin remodeling genes accounted for 16% (13 of 80) of the chordoma tumors we sequenced (Fig. 2). Genomic alterations involving the *CDKN2A/B* loci were the next prevalent event (*CDKN2A*/+, 13.8% patients, nine with 9p21.3 homozygous deletion, one with SVs involving

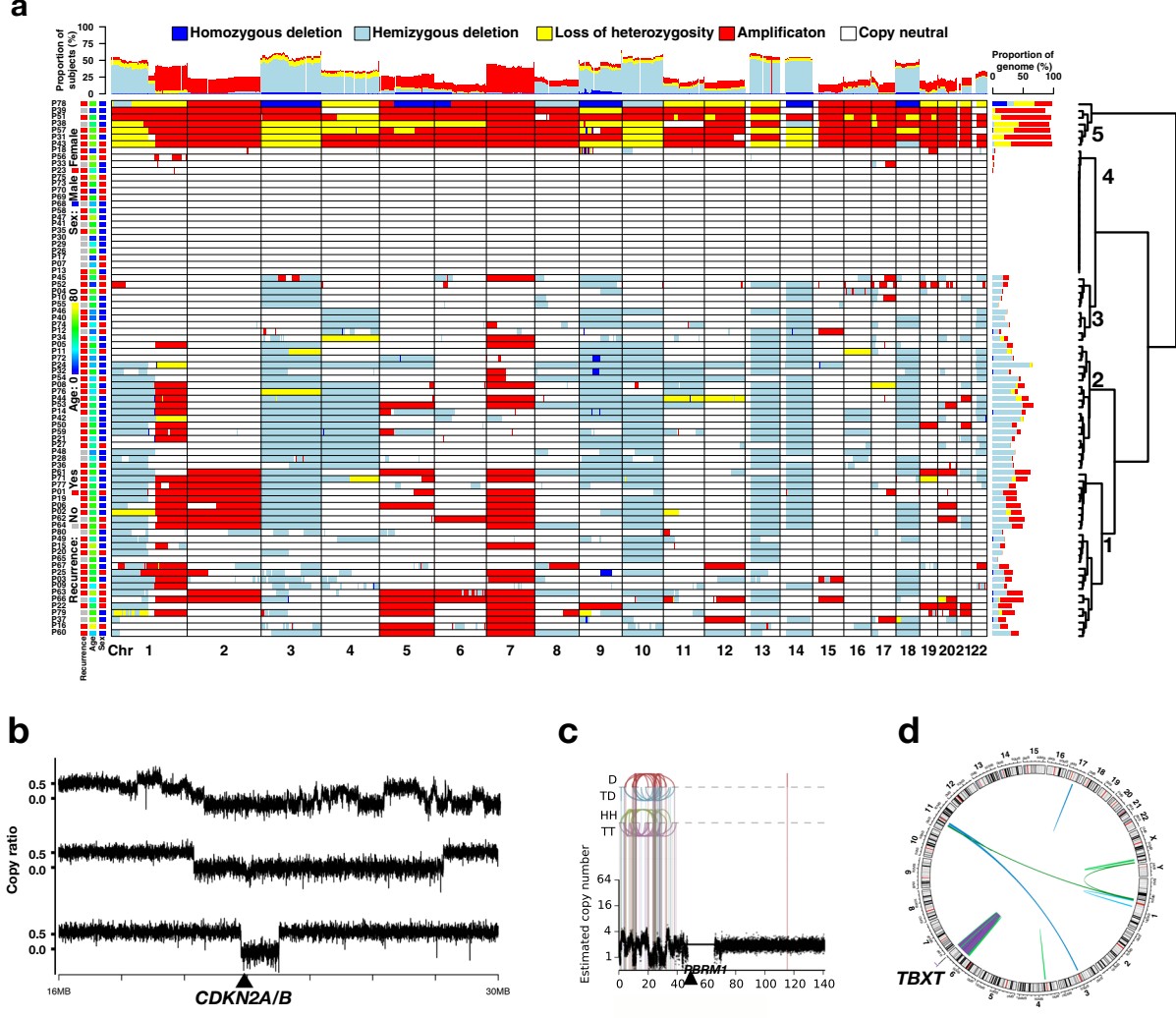

**Fig. 3 Somatic copy number alterations (SCNAs) and structural variants (SVs) in skull-base chordoma patients. a** Genome-wide SCNAs. Patients were clustered into five groups. Group 1: extensive SCNAs, lacking deletions of 4, 9, and 14; Group 2: extensive SCNAs, lacking amplifications; Group 3: scattered SCNAs, most of which are deletions; Group 4: no or few SCNAs; and Group 5: extensive amplifications demonstrating characteristic consistent with whole-genome doubling. **b** Representative examples showing complex rearrangements at chromosome 9p. Plots show copy ratios (tumor vs. matched normal) of the chromosome 9p21 region. All three samples show an arm-level deletion coupled with complex rearrangements (top) and focal deletions of the 9p21.3 region (middle) or the *CDKN2A/2B* locus (bottom), resulting in the homozygous loss of *CDKN2A/2B*. **c** An example of complex rearrangements at chromosome 3p. D deletion, TD tandem duplication, HH head-to-head inverted, TT tail-to-tail inverted. **d** A chordoma sample showing chromothripsis of the 6q region, where the *TBXT* gene is located. See also Supplementary Tables 4 and 5.

the *CDKN2A/2B* loci, and one with *CDKN2A* mutation), which occurred mutually exclusively with *PBRM1* and *SETD2* alterations (Fig. 2). Genomic alterations disrupting the *TBXT* gene (*TBX*+) were only observed in four patients, including a patient with germline *TBXT* duplication, two tumors with somatic focal *TBXT* duplication, and one tumor with 6q chromothripsis. Disruptions of *LYST* (*LYS*+) were seen in three tumors; one of them also had a *PBRM1* mutation and another had *TBXT* germline duplication.

In summary, we identified candidate driver events (*PBRM1*+, *SETD2*+, *CDKN2A/B*+, *TBXT/LYST*+) in 33.75% (27 out of 80) of the skull-base chordoma patients we sequenced. The remaining tumors might be caused by nonsynonymous mutations or SCNAs/SVs in known cancer driver genes (observed in 8% patients, Fig. 2) or other driver genes/mechanisms for which statistical power was too low to detect in this study. In particular,

some of these tumors showed extensive chromosomal SCNAs that are consistent with chromosomal aneuploidy (Supplementary Fig. 7), which may drive chordoma development, similar to what has been previously reported in chromophobe renal cell carcinomas and pancreatic neuroendocrine tumors that also showed high fractions of tumors without known drivers[25].

*PBRM1*+ tumors had higher total mutation (Wilcoxon rank sum, $p = 0.04$) and nonsynonymous mutation burden ($p = 0.07$), were more likely to be in SCNA group 1 (reference: SCNA Groups 3 and 4; $p = 0.02$), had higher fractions of *APOBEC* mutational signatures ($p = 0.09$), and higher number of SVs, particularly tandem duplications ($p < 0.0001$) and inter-chromosome translocations ($p = 0.03$), as compared to tumors without *PBRM1* alterations (*PBRM1*−). *PBRM1*+ tumors had reduced expression of *PBRM1* ($p = 0.16$) compared to *PBRM1*− tumors, although the difference was not statistically significant

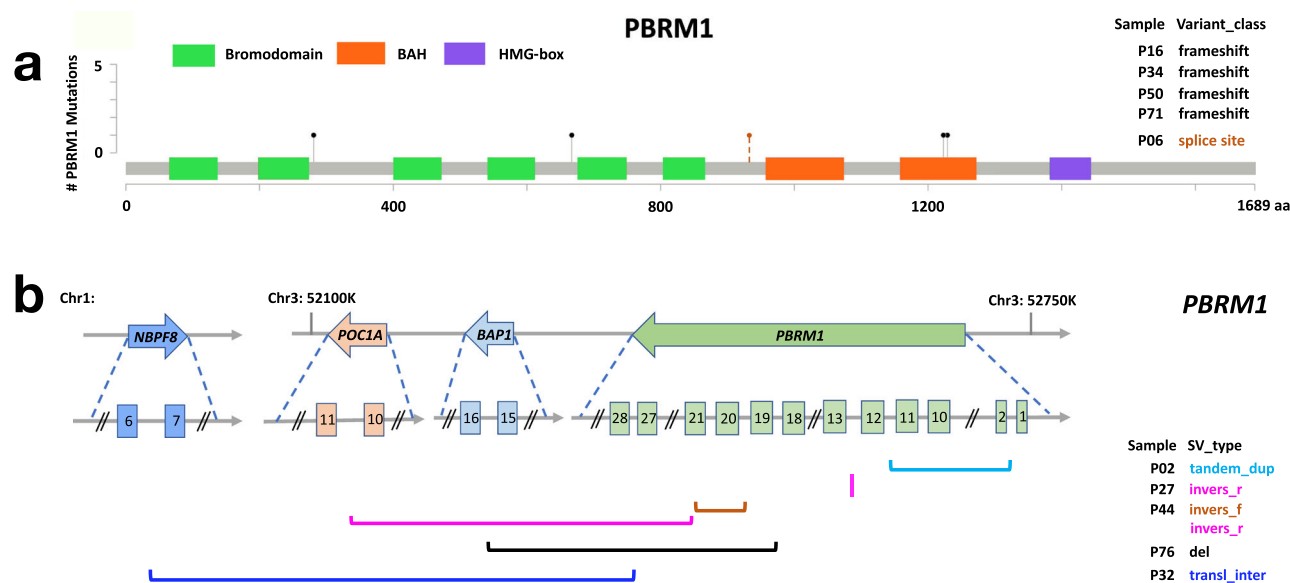

**Fig. 4 Summary of *PBRM1* alterations identified in ten skull-base chordoma patients. a** Lollipop plots depict nonsynonymous mutations found in *PBRM1* (five patients), showing identified mutations relative to a schematic representation of the gene. Each lollipop denotes a unique mutation and the length of the line reflects the number of samples with the mutation. The colored boxes are functional domains of the protein. The splicing mutation, which is located in a noncoding region, is shown with a broken line. **b** Structural variants (SVs) found in *PBRM1* (five patients) detected by Meerkat. Different SV types are indicated by different color schemes.

(Supplementary Fig. 8A). Similarly, *CDKN2A/B+* tumors had lower *CDKN2A* expression level ($p = 0.003$, Supplementary Fig. 8B) and they were also more likely to have tumors with higher nonsynonymous mutation burden ($p = 0.002$). Associations between mutation burden and alterations in *PBRM1* ($p = 0.12$) and *CDKN2A/2B* ($p = 0.06$) were similar but less significant (possibly due to smaller sample size) when restricting to patients without presurgery RT.

**Somatic driver landscape in relation to patient characteristics.** Among the 80 distinct patients with WGS data, 64 had conventional/classical, 14 had chondroid, and 2 had dedifferentiated chordoma. Mutational burden, total numbers of SVs, and driver genomic events (*PBRM1* and *CDKN2A/B*) did not vary significantly between conventional and chondroid chordoma tumors. Both patients with dedifferentiated chordoma had 9p21 homozygous deletions as well as high mutational burden and complex structural alterations. One of them was diagnosed with chordoma at age 9 years old and her tumor recurred 6 months after surgery. In contrast, the other nine pediatric chordoma patients (diagnosed ≤ 20 years) appeared to have quiet genomes, characterized by low mutational burden and the absence of driver events (alterations in *PBRM1*, *CDKN2A/B*, *TBXT* genes), compared with adult chordoma patients.

**Somatic driver landscape and clinical prognosis.** As it is clinically important and challenging to identify chordoma patients with aggressive features, we sought to identify genomic alterations that were associated with patient outcomes (17 deaths and 59 recurrences). We first examined the two potential driver events, *PBRM1* and *CDKN2A/B* status, in relation to chordoma-specific survival (CSS) and recurrence-free survival (RFS). After adjustment for age, sex, presurgery and post-surgery RT, *PBRM1* alterations were significantly associated with worse CSS (hazard ratio (HR) = 4.79, 95% confidence interval (CI) = 1.57–14.59, $p = 0.0058$) and RFS (HR = 5.72, 95% CI = 2.68–12.19, $p = 6.4 \times 10^{-6}$) (Fig. 5).

In contrast, *CDKN2A/B+* status (9p21.3 homozygous deletion, SVs involving the *CDKN2A/2B* loci, and *CDKN2A* mutation) was not significantly associated with CCS (HR = 0.88, 95% CI = 0.20–3.92, $p = 0.86$) or RFS (HR = 1.68, 95% CI = 0.66–4.31, $p = 0.28$). However, when we looked at 9p and 9q arm-level deletions and focal SCNAs on chromosome 9, we found that arm-level 9q deletion and focal deletions of 9p11.2, 9p21.3, and 9q21.11 were significantly associated with RFS (Fig. 5 and Supplementary Table 6). Interestingly, the RFS association was stronger for 9q21.11 focal deletion (HR = 3.63, 95% CI = 1.97–6.69, $p = 3.44 \times 10^{-5}$) as compared to deletions of the 9p21.3 region, where *CDKN2A/2B* is located (HR = 2.65, 95% CI = 1.42–4.93, $p = 0.002$). These findings suggest that the RFS association was not entirely driven by *CDKN2A/2B* alterations.

Next, we tested the associations between 17 significant arm-level SCNA events and clinical outcomes. After multiple testing correction, we found that 22q deletion was significantly associated with CSS (HR = 5.88, 95% CI = 1.85–18.68, nominal $p = 0.0027$, Bonferroni corrected $p = 0.046$) and RFS (HR = 3.74, 95% CI = 1.89–7.38, nominal $p = 0.00014$, Bonferroni corrected $p = 0.0024$) (Supplementary Table 6). Chromosome 22q harbors an important SWI/SNF gene, *SMARCB1/IN1*, and the complete loss of SMARCB1 expression on immunohistochemistry due to homozygous *SMARCB1* deletion has been used as a marker for a rare chordoma subtype, poorly differentiated chordoma[8,26]. Homozygous deletion of *SMARCB1* was not seen in our patients. However, we found that RNA expression of *SMARCB1* was significantly lower in tumors with arm-level chromosome 22q deletion compared to those without the deletion in a subset of patients with RNA-Seq data available ($p = 0.001$, Supplementary Fig. 8C).

When we combined the two significant events for CSS (*PBRM1* and 22q deletion), the association for CSS became slightly stronger (HR = 10.55, 95% CI = 2.81–39.64, $p = 0.001$). Similarly, the combination of *PBRM1*, 9q21.11 deletion, and 22q deletion also improved the association for RFS (HR = 4.22, 95% CI = 2.34–7.62, $p = 1.77 \times 10^{-6}$; comparing harboring any event to no event). HRs for the associations between these genomic

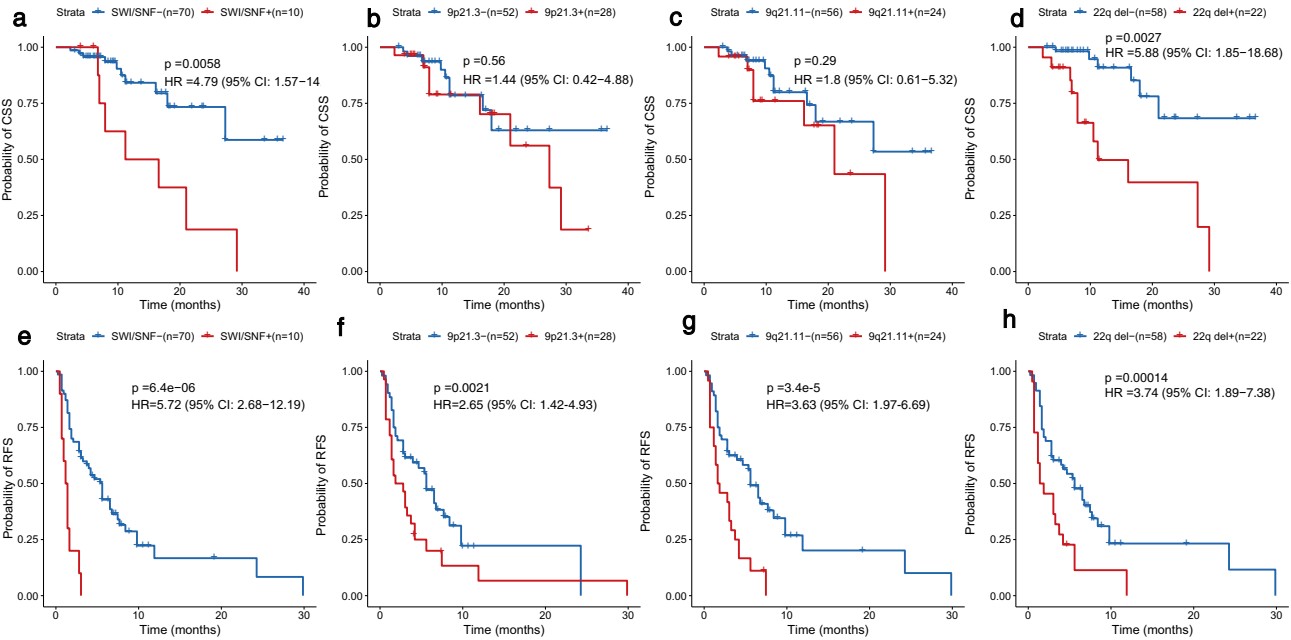

**Fig. 5 Genomic features in relation to chordoma-specific survival (CSS) and recurrence-free survival (RFS).** CSS (**a–d**) and RFS (**e–h**) of chordoma patients stratified by *PBRM1* status, 9p21.3 deletion, 9q21.11 deletion, and 22q deletion. p values, hazard ratios (HRs), and 95% confidence intervals (CIs) were obtained from Cox proportional hazards models with the adjustment of age, sex, presurgery and post-surgery radiation therapy. "+" indicates subjects carrying the genomic alteration. See also Supplementary Table 6.

features and CSS/RFS did not change significantly with the additional adjustment of tumor KI67 status, gross resection rate, TMB, and SCNA group (Supplementary Table 6C). Associations for *PBRM1* alterations and 22q deletion for both CSS and RFS remain significant when we restricted the analyses to the 53 patients who were not previously diagnosed with or treated for chordoma and who did not have presurgery RT (Supplementary Table 6C).

**Paired primary and recurrence/metastasis comparison.** We sequenced the genome of 11 paired primary and local recurrent tumor samples with the same sequencing depth. On average, recurrence samples showed increased number of SNVs (30.1% higher), indels (43.1% higher), genomic regions covered by SCNAs (2.1% higher), and SVs (43.5% higher), compared with the matched primary tumor samples. The time to the first recurrence (TTFR) ranged from 3 to 36 months with a median of 8 months. As expected, TTFR was positively associated with the number of SNVs (Pearson correlation $(r) = 0.47$, $p = 0.14$) and indels $(r = 0.64$, $p = 0.035)$ that were specific to recurrence samples (acquired during tumor progression). However, recurrent events, especially those involving known cancer driver genes, that were specific to either primary or recurrence samples were rarely observed in more than one patient.

Overall, the paired primary and recurrent tumor samples showed high numbers and proportions of shared SCNAs and, to a lesser extent, SNVs, indels (Fig. 6a, d), suggesting that SCNAs are fundamental to the initiation of chordoma. Mutational signatures and SV events also showed similar patterns between paired primary and recurrence samples for most patients (Fig. 6b, c, e).

Independently from the primary analysis of 80 patients, we analyzed a patient with metastatic chordoma and compared genomic profiles among the chordoma recurrence (R), LM, and TM samples. The total number of mutations was 1564, 1595, and 1552, for R, LM, and TM samples, respectively. Mutations in known cancer driver genes were not found in any of these

samples, therefore, the mutation evolution analysis was not informative. Focal deletion of the *CDKN2A* region, which was seen in all three samples, is likely the driver event for this patient (Supplementary Fig. 9). Overall, the SV/SCNA profile and mutation signatures were very similar across the three samples (Fig. 6a–c, h). Similarly, the three samples also shared high proportions of SNVs and indels (Fig. 6f, g), suggesting a monoclonal origin. Interestingly, *APOBEC* signatures (SBS2 and SBS13) were only present in the TM sample, while absent in R and LM samples, which is consistent with results from a recent study reporting that *APOBEC* mutagenesis could generate mutations late in the evolution of metastatic disease in thoracic tumors[27].

## Discussion

In this genomic analysis of skull-base chordoma, we described a comprehensive genomic landscape of this rare cancer and identified several genomic features that were associated with disease progression and outcomes.

We showed that skull-base chordoma was among the cancer types that had the lowest mutation rates, similar to pediatric brain tumors and leukemias[28]. Driver gene analysis only detected one significantly mutated driver gene in this patient cohort, *PBRM1*. In addition, chromosome 3p21, where *PBRM1* and *SETD2* are located, is the region with the most prevalent chromothripsis and one of the most frequently deleted regions. Given the role of *PBRM1* and *SETD2* in chromatin remodeling, our data suggest that epigenetic dysregulation may play an important role in chordoma development. In line with this, previous Pan-cancer analysis found that significant focal SCNA regions without known cancer genes were enriched with genes involved in epigenetic regulation[29]. In our study, we identified several significant focal SCNA regions that did not contain known cancer driver genes. Future mechanistic studies should follow-up genes in these focal SCNA regions to identify additional driver genes/mechanisms.

Consistent with findings from previous studies of sacral chordoma[9,19,20,30], we found that alterations of *PBRM1* and

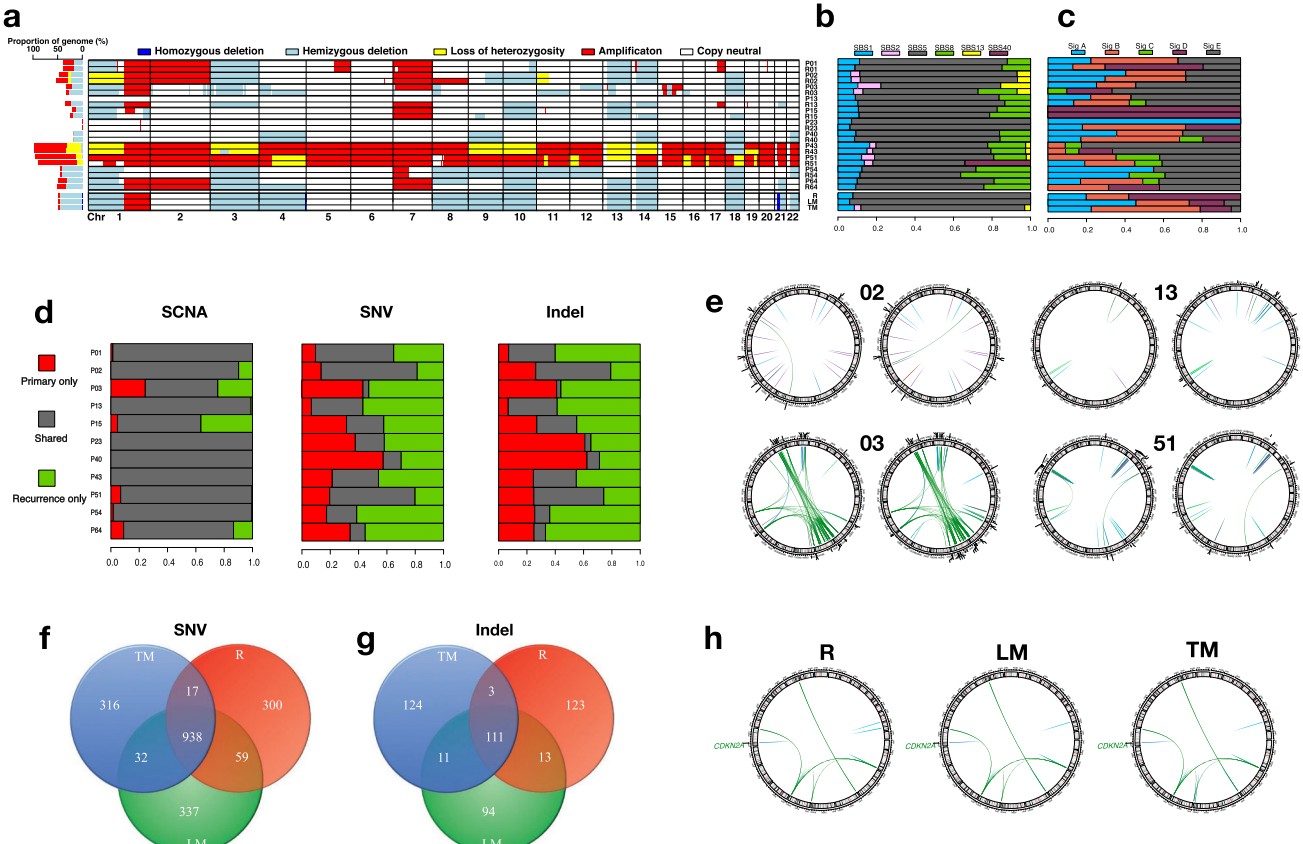

**Fig. 6 Comparison of genomic alterations between paired primary tumor and recurrence/metastasis samples.** Comparison of somatic copy number alterations (SCNAs, panel (**a**)), COSMIC single-base substitution (SBS) signatures (panel (**b**)), and de novo indel signatures (panel (**c**)) in 11 paired primary tumor (P) and recurrence (R) samples (upper) as well as three paired tumor and metastasis samples (lower). R recurrence, LM lymph node metastasis, TM thoracic metastasis. **d** Shared proportion of SCNAs, single nucleotide variations (SNVs), and indels in 11 paired primary tumor and recurrence samples. Each bar indicates the proportion of alterations in primary tumor only (red), recurrence only (green), and shared between primary and recurrence samples (gray) within each patient. **e** Comparison of structural variants (SVs) in four paired primary tumor and recurrence samples with total number of SVs larger than 20. Number of SNVs (**f**) and indels (**g**) shared among R, LM, and TM samples. **h** Comparison of SVs among R, LM, and TM samples. See also Supplementary Fig. 8.

*CDKN2A/2B* locus were the most common events in chordoma. In addition, our results suggest that they occurred in a mutually exclusive manner. However, somatic duplications of *TBXT* and mutations in PI3K signaling genes were rare in our study, which differs from what was observed in sacral chordoma[9]. The discrepancies might be due to the differences in chordoma site (skull base vs. sacral), sequencing platform (WGS vs. primarily targeted), or study populations (East Asian vs. European). Higher frequencies of *TBXT* amplifications reported previously might be attributed to the use of higher-resolution technologies, such as high-depth sequencing and FISH, to detect low-copy gains. In addition, variations in how SCNAs are defined (arm/chromosome level vs. focal) may also cause discrepancies. In our study, although one-copy 6q gain was observed in nine samples (11.25%), focal *TBXT* amplification was only seen in two patients. Consistent with our findings, a genome-wide SNP genotyping array analysis of skull-base chordoma also reported low frequency of focal *TBXT* amplification (1 of 18 patients)[20], suggesting that focal *TBXT* amplifications may be more common in sacral than skull-base chordomas. Given the low incidence as well as low mutation rates of chordoma, collaborations across different research groups such as chordoma foundation for a large-scale chordoma genomic analysis in diverse populations are needed to delineate the genomic landscape of this rare disease.

Interestingly, the comparison of mutation signature profiles between chordoma and other cancer types suggests that chordoma is most similar to kidney cancers and bone cancer. The morphological overlap between chordoma and ccRCC is known[31] but the mechanism underlying this similarity remains unclear. Like chordoma, chromosome 3p loss and *PBRM1* mutations are very common in sporadic ccRCCs[32,33]. Our findings of the similar somatic driver genes and mutation signatures in chordoma and kidney cancers suggest that there might be common etiologic factors associated with these two cancer types and identifying these factors may provide etiologic insights for chordoma.

Recently, in a WES/WGS study of 11 chordoma patients, Gröschel et al. demonstrated that SBS3, a mutational signature associated with homologous recombination deficiency (HRD) and *BRCA1/2* germline mutations, was significantly enriched in advanced chordoma samples, thus implicating the poly(ADP-ribose) polymerase (PARP) inhibitor as a promising therapeutic option[34]. However, SBS3 was not present in any of the patients we sequenced. In addition, no pathogenic/LP germline variants in *BRCA1/2* genes were found in our patients. Using HRDetect scores, which summarized six mutation features associated with HRD (SBS3, SBS8, SV signature 3, SV signature 5, HRD index, and the fractions of deletions with micro-homology)[35], we

identified six tumors with suggestive HRD signature, each of which was primarily driven by a single feature. None of the six tumors showed consistent HRD scores across different features, which was a characteristic of HRD observed in breast cancer. In the study by Gröschel et al., all patients had progressive disease and had previously received RT or systemic treatment prior to surgical resection, whereas few patients in our study had metastases and only a small number of patients received RT prior to surgery. Our results did not support defective HR as a common mechanism in skull-base chordoma patients.

The phylogenetic analysis was not informative due to low mutation burden and lack of driver mutations in recurrent/metastatic tumors. However, by comparing genomes in paired primary and recurrence/metastasis samples, we found that paired samples were clonally related for all patients and the largest shared fractions between the paired samples were observed for SCNAs compared to SNVs and indels, suggesting the importance of SCNAs in tumor initiation. Nevertheless, we did find APOBEC signatures were present in distant metastatic sample but absent in recurrent tumor and lymph nodes, supporting views from recent studies that APOBEC mutations may be switched on at various stages of tumor evolution[27].

Clinical outcomes in skull-base chordoma patients are highly variable and disease progression is likely determined by both surgical factors and tumor biology. Currently, there is no clear clinical guidance on patient stratification regarding treatment such as post-surgery RT. Multiple markers have been proposed, but most were based on candidate marker searches in small studies. In our study, we examined a comprehensive list of genomic alterations (including mutations, SCNAs, and SVs) and their associations with clinical outcomes with adjustment of all potential confounders. Our results suggest that PBRM1 alterations were one of the most significant prognostic factors for skull-base chordoma, which is consistent with previously reported associations between PBRM1 mutations and late stage and poor prognosis in ccRCC[32,33]. Notably, recent clinical data showed that PBRM1 inactivation may predict benefit from anti-PD-1 checkpoint inhibitors in ccRCC[36] and Pbrm1-deficient murine melanomas were more strongly infiltrated by cytotoxic T cells[37], suggesting that immune checkpoint inhibitors have the potential of treating chordoma patients with PBRM1 alterations.

We also found that the deletion of chromosome 22q, where SMARCB1 is located, was significantly associated with CSS and RFS. The complete loss of SMARCB1 expression is considered a hallmark of poorly differentiated chordoma, a rare and aggressive chordoma subtype[8,26,38,39]. Although the homozygous loss of SMARCB1 was not seen in our chordoma patients, patients with hemizygous 22q deletion showed reduced expression of SMARCB1 at the mRNA level. Our results suggest that partial inactivation of this gene was associated with worse patient outcomes and might be used as a prognostic marker in conventional chordoma. These findings further demonstrate the importance of SWI/SNF complex genes in chordoma initiation and progression, highlighting the need for examining SWI/SNF genes in chordoma clinics.

Consistent with previous reports, we found that 9p21 deletion was associated with worse RFS. However, complex genomic rearrangements occur on chromosome 9, including arm-level and multiple focal deletions involving both 9p and 9q. These regions showed high co-occurrence and most were associated with RFS. Surprisingly, the stronger association was observed for 9q deletion, especially the focal 9q21.11 deletion, compared to arm-level 9p or 9p21.3 focal deletion. In particular, CDKN2A/2B alterations (primarily 9p homozygous deletions) were not associated with RFS. These results suggest that the RFS association is unlikely to be entirely driven by CDKN2A inactivation. The 9q21.11 deletion

peak contains hsa-mir-1299, LOC440896, PGM5P2, and FOXD4L6. Although the biological relevance remains unclear, this region warrants further investigations as a marker for recurrence in skull-base chordoma.

Remarkably, when we combined PBRM1 alterations and 22q deletion, the associations for both CSS and RFS became stronger, demonstrating the potential of designing a multi-marker panel in CSS prediction. Future studies with large number of patients to validate these markers in relation to disease outcomes are warranted.

Our study is still relatively small, which may have limited our ability to detect additional driver genes and de novo mutation signatures. Since RT was performed at multiple institutes/clinics, the detailed data on RT type, duration, and dosage were not available. In addition, the lack of multiple specimens from a single patient in combination with the low mutation burden has made phylogenetic analysis to study tumor evolution challenging. Despite the limitations, our study was derived from a clinically well-annotated patient population. Unlike most previous genomic studies in which advanced tumors were usually enriched, most patients in our study did not receive extensive therapies prior to surgeries and therefore the genomic profiles should not have been heavily influenced by treatment. Our analysis provided a detailed genomic landscape of skull-base chordoma, which may have important clinical implications in patient stratification and targeted treatments.

## Methods

In this study, we analyzed data and biospecimens from patients who were diagnosed with skull-base chordoma and underwent endoscopic endonasal surgeries at the Neurosurgery Department of Beijing Tiantan Hospital, Capital Medical University, between October, 2010 and November, 2017. Fresh-frozen primary tumor, recurrent tumor whenever possible, and matched peripheral blood samples were collected from these patients. Clinicopathological characteristics including age, tumor histologic type, tumor volume, Ki67 status, gross resection rate, presurgery RT, post-surgery RT, recurrence, and death status were recorded using Microsoft Excel 2016 (v16.0). The chordoma diagnosis was confirmed with brachyury staining for 70 patients and was confirmed by morphology in combination with CK and EMA markers for the remaining ten patients, as illustrated in Supplementary Fig. 10 for the two patients with IDH1 mutations. The study protocol was approved by the ethics committee of the Beijing Tiantan Hospital and the written informed consent was obtained for all study participants.

**Biospecimen collection, quality control, and processing**. A small section of frozen tumor samples was embedded in optimal cutting temperature compound, frozen with cold N-hexane, and then cut to 10-μm-thick frozen sections using cryostats (Leica, Germany). The frozen chordoma tissue sections were fixed in cold acetone, stained with hematoxylin and eosin, and then dehydrated through increasing concentrations of ethanol and xylene. Only tumors with >50% tumor cells were included for DNA/RNA extraction.

**DNA and RNA extractions**. Genomic DNA was extracted from frozen tissue specimens and matched peripheral blood, respectively, using DNeasy blood & tissue kit (Qiagen, CA). A total of 500 ng DNA with high-molecular weight (>20 Kb single band) was used for the library preparation. For total RNA extraction, tissue sections were processed with TRIzol (Thermo, USA) according to the manufacture instructions. RNA was run on 1% agarose gels to check for degradation and contamination. RNA quality and quantity were assessed using the RNA Nano 6000 Assay Kit of the Bioanalyzer 2100 system (Aligent Technologies, CA, USA). RNA samples with an integrity number of over 6.8 were included for transcriptome library preparation and sequencing.

**Library construction, sequencing, and data generation**. WGS and RNA-Seq were carried out by the Novogene Corporation (Beijing, China). The WGS library was constructed using Truseq Nano DNA HT Sample Prep Kit (Illumina, USA) and sequenced on Illumina HiSeq X platform with the average depth of 76X for tumors and 41X for matched germline. After the exclusion of reads containing adapter contamination and low-quality/unrecognizable nucleotides, the clean data were mapped to the reference human genome (UCSC hg19) using the Burrows–Wheeler Aligner software (VN:0.7.8-r455, http://bio-bwa.sourceforge.net/)[40] to get the original mapping results stored in the BAM format. SAMtools (v1.8, http://samtools.sourceforge.net/)[41], Picard(v2.18.20, http://broadinstitute.github.io/picard/), and GATK (v3.8-1-0, http://software.broadinstitute.org/gatk)[42] were used to sort BAM

files and to do base quality recalibration, duplicate reads removal, and local realignment to generate final BAM files for mutation calling. BAM-matcher (2016 version)[43] was used to verify whether two BAM files (tumor/tumor pair or tumor/normal pair) were generated from the same patient.

**Somatic variant calling**. To detect somatic SNVs, five somatic callers (Strelka (v2.7.1), https://github.com/Illumina/strelka)[44], Sentieon TNsnv (https://support.sentieon.com/appnotes/out_fields/) and TNhaplotyper (https://support.sentieon.com/appnotes/out_fields/), which are commercial versions of MuTect and MuTect2[45], Lofreq (V2.1.3.1, https://csb5.github.io/lofreq/)[46] and MuSE (V1.0rc, https://bioinformatics.mdanderson.org/public-software/muse/)[47] were applied to all tumor/normal BAMs. SomaticSeq (v2.7.2, https://github.com/bioinform/somaticseq)[48] was then applied to combine the variants called from these five callers to generate the final ensemble results. Only the variants called by two or more callers were retained. For indels, three of these callers were applied (Strelka, TNhaplotyper, and Lofreq) for indel calling and variants called by two or more callers were included in subsequent analyses. Variants were excluded if they did not pass the pipeline quality control metrics, had variant allele fraction (VAF) < 0.07 in tumor, VAF > 0.02 in normal, alternative allele read count < 3 or total read count < 8 in tumor, total read count < 6 in normal, and if the minor allele frequency (MAF) was >0.1% in 1000 Genomes Project[49], the ESP6500 data set from University of Washington's Exome Sequencing Project (http://evs.gs.washington.edu/EVS/) or ExAC[50]. Manual review of variants in several genes was performed using the Integrative Genomics Viewer (v2.3.61)[51].

TMB was defined as the number of somatic mutations in the coding region per megabase, including SNVs and indels. When comparing TMB in our chordoma tumors to other tumor sequenced in TCGA, we used normalized TMB, for which the distribution of variants was compiled from over 10,000 WGS samples across 33 TCGA landmark cohorts using maftools (v1.6.05)[52].

**Potential drivers**. dNdScv[17] was used to identify driver genes, and *PBRM1* was the only significantly mutated gene identified in our patient cohort (FDR < 0.01). Since *PBRM1* is a SWI/SNF gene and alterations in SWI/SNF genes have been suggested to play potential driver roles in sacral chordoma, we consider genomic alterations in SWI/SNF genes, among which only alterations in *PBRM1* and *SETD2* (both located at chromosome 3p21.3) were identified in this patient cohort, as potential driver events. These included mutations in *PBRM1* and *SETD2*, structural variants involving the two genes, and significant chromosome 3p chromothripsis regions occurring at 3p21.3, all of which were validated by targeted sequencing. In addition, we also considered homozygous deletion of the *CDKN2A/2B* locus and mutations in previously reported chordoma genes *T* and *LYST*[9] as potential driver events. To identify driver genes or recurrently mutated genes, we focused on nonsynonymous mutations in the coding region.

**Mutation validation**. We selected nonsynonymous mutations in 36 genes identified by WGS, including *PBRM1, SETD2, CDKN2A/2B, MAP3K4, BAG1, ITGA6, CSDE1*, etc., and validated these mutations using Sanger sequencing.

**Mutation signature analysis**. We performed mutational signature analysis for SBS, DBS, and small indels (ID) using SigProfiler (Alexandrov et al.[53]) (v0.0.5.77, https://github.com/AlexandrovLab), as previously described in original publications[12]. We first performed de novo mutation signature analysis. For each extracted mutation signature, we identified a set of COSMIC signatures whose linear combinations best approximated the given de novo mutation signature. Cosine similarity index was calculated between the de novo mutation signature and the linear combination of COSMIC signatures to evaluate whether the de novo signature was novel. We then used SigProfiler to evaluate the contribution of each COSMIC signature for each tumor sample.

PyClone (v0.13.1, http://compbio.bccrc.ca/software/pyclone)[14] was used to classify SNVs into clonal and subclonal SNVs, accounting for CN and purity estimated by FACETS (v0.5.6, https://github.com/mskcc/facets)[18]. We then performed mutation signature analysis separately for clonal and subclonal mutations to investigate mutation signatures associated with tumor initiation and progression.

**Somatic copy number alteration (SCNA) analysis**. Allele-specific SCNA analysis was performed using FACETS (version 0.5.6, https://github.com/mskcc/facets)[18]. An arm-level SCNA was defined if the SCNA event covered 90% of the p or q arm of each chromosome. GISTIC2.0[21] was used to detect significantly mutated regions. In addition, FACETS also estimated clonal status of each SCNA event based on the distribution of VAF. In our data set, 89.5% of SCNA events were estimated to be clonal. We performed GISTIC analysis separately on clonal SCNAs and subclonal SCNAs, but we did not identify additional significantly mutated regions that were specific to clonal or subclonal SCNAs.

We also used FACETS to estimate purity for samples with sufficiently informative SCNA events. For samples without informative SCNA events, we estimated purity based on SNVs. Briefly, we identified SNVs located in copy neutral regions based on FACETS and assumed the number of the mutant allele followed a mixture of binomial distributions $x_i \sim \sum_{k=1}^{K} \pi_i \mathrm{Binom}(x_i; N_i, \theta_i)$ where $\theta_K < \ldots < \theta_1 \leq 0.5$.

Parameters were estimated using the expectation–maximization algorithm, and the number of clones was determined by the Bayesian Information Criterion. The purity was estimated as $2\theta_1$, because we used only SNVs in copy neutral regions. For samples with purity estimated successfully by both methods, the purity estimates were highly concordant with Pearson correlation coefficient = 80.4%.

Hierarchical clustering was performed using a function *hclust* based on SCNA profiles weighted by the length of each SCNA event. At each base pair, the distance between two tumors was defined as 0 if they had the same CN status (homozygous deletion, hemizygous deletion, copy neutral, amplification, loss of heterozygosity) and 1 otherwise. The number of clusters was determined by the Elbow method.

The allele-specific SCNA status inferred based on FACETS was used to determine the status of WGD. Specially, a tumor was considered to have undergone WGD if more than 50% of the autosomal genome had an MCN (CN of the major allele, the more frequent allele in a given segment) ≥2[22].

**Structural variant (SV) analysis**. We used the Meerkat (v0.189, http://compbio.med.harvard.edu/Meerkat/) algorithm[23] to call somatic SVs and estimate the corresponding genomic positions of breakpoints from recalibrated BAM files in the primary analysis. We used parameters adapted to the sequencing depth for both tumor and normal tissue samples and the library insert size. To compare results, we also called SVs with our newly developed pipeline (MoCCA-SV, v0.2, https://github.com/NCI-CGR/MoCCA-SV)[54] that integrates results from four callers, Svaba (v1.1.0)[55], Breakdancer (v1.4.5)[56], Delly (v0.8.1)[57], and Manta (v1.4.0)[58]. The criteria for comparison across callers were 70% reciprocal overlap and 50 bp window for intrachromosomal breakends. We focused on events called by all four callers or ≥3 callers for SV candidates already found in one sample by Meerkat.

**Chromothripsis analysis**. We used ShatterSeek (https://github.com/parklab/ShatterSeek) to identify chromothripsis events[24]. Permutation tests were performed to identify such rearrangement clusters followed by manual curation for all cases. Only regions with high interactions were included. Regions affected by chromothripsis were characterized by clusters of breakpoints belonging to SVs that are interleaved. Thus, high-confidence chromothripsis events were identified if a region satisfied one of the following sets of statistical criteria: (1) ≥6 interleaved intrachromosomal SVs, seven adjacent segments oscillating between two CN states, the fragment joins test $p < 0.05$, and either the chromosomal enrichment or the exponential distribution of breakpoints test $p < 0.05$; and (2) at least three interleaved intrachromosomal SVs and four or more interchromosomal SVs, seven adjacent segments oscillating between two CN states and the fragment joins test $p < 0.05$.

**Pathogenicity scoring of germline rare variants**. Three variant callers, GATK (v3.8-1-0) HaplotyperCaller, UnifiedGenotyper, and FreeBayes (v1.2.0)[59] were used to call germline variants. The GATK LeftAlignAndTrimVariants module was also applied for the normalization of the variants obtained from the three callers. The majority-voting ensemble approach with bcbio (https://github.com/bcbio/bcbio.variation.recall) was applied to combine normalized variants from three callers. The variants called by two or more callers were used for further analyses.

Rare variants (MAF < 0.001 in ExAC East Asian population) passing quality control and variant filters were evaluated for pathogenicity. A step-wise pipeline was constructed to evaluate each rare variant in the genes of interest, as previously described (Mirabello, under revision). Briefly, variants were classified as "Pathogenic" (P), "Likely Pathogenic," "Variant of Uncertain Significance" (VUS), "Likely Benign," or "Benign" based on ClinVar (26), InterVar version 2.1.2 (default settings)[60], impact (frameshift indels, stop gain/loss, or known splice sites), HGMD (2018.1)[61], and manual review of the published literature. All P and LP designated variants, the high impact and HGMD DM variants were manually reviewed for a final designation of P, LP, or VUS_D.

**Germline duplication of *TBXT***. PennCNV-seq software (v1.0.4, https://github.com/WGLab/PennCNV-Seq)[62] was performed to call germline copy number variations (CNVs) on 80 normal samples. Using BAM files and the reference genome (FASTA file), sequence counts and B allele frequencies (BAF) were generated. SAMtools was used to calculate the coverage (with mpileup) and log R ratios (LRR). Population frequency of the B allele for each marker was generated. Hidden Markov model in the package was used to detect potential germline CNVs and generate CNV calls. Adjacent CNV calls were merged. We removed CNV calls from samples with LRR_SD ≥ 0.35, calls from centromeric regions, and calls with number of SNPs < 10 or length < 1000 bp. CNV calls from region chr6:160,000,000 to 170,000,000, where the *TBXT* gene is located, were extracted. Plots for calls with this region were generated and reviewed. The presence of *TBXT* duplication was confirmed using an independent analysis, the Canvas (v1.31) CNV caller developed by Illumina[63]. Briefly, Canvas scans for genomic regions with statistically significant different number of short read alignments, with the baseline estimated assuming most of the genome is diploid and reads are distributed randomly across the genome. Regions with fewer than the expected number of alignments are classified as losses. Regions with more than the expected number of alignments are classified as gains. Germline duplications covering the *TBXT* gene were visually verified by plotting the LRR and the BAF in the genomic region.

**RNA sequencing**. A total amount of 3 μg RNA per sample was used as input material for RNA sample preparations. First, we removed ribosomal RNA using the Epicentre Ribo-zeroTM rRNA Removal Kit (Epicentre, USA). Subsequently, sequencing libraries were generated using the rRNA-depleted RNA and NEBNext® UltraTM Directional RNA Library Prep Kit from Illumina® (NEB, USA) following manufacturer's recommendations. After adapter ligation and library amplification, the library fragments were purified with AMPure XP system (Beckman Coulter, Beverly, USA) in order to select fragments of preferentially 150–200 bp in length. The strand marked with dUTP was not amplified, allowing strand-specific sequencing. Finally, products were purified (AMPure XP system) and library quality was assessed on the Agilent Bioanalyzer 2100 system. After cluster generation, the libraries were sequenced on an Illumina Hiseq platform and 150 bp paired-end reads were generated. Raw data of fastq format were processed and clean data were generated by removing reads containing adapters or ploy-N and low-quality reads. Gene expression was quantified as TPM (transcript per million) using RSEM (https://github.com/deweylab/RSEM)[64], and log$_2$TPM was used for statistical analyses.

**Statistical analyses**. The Wilcoxon rank test was used to compare mean differences in genomic alterations across different groups of patients stratified by treatment, clinical or genomic features. Multivariate regression analysis was used to assess the associations between multiple genomic features and patient characteristics, with the adjustment of age at diagnosis, sex, presurgery and post-surgery RT. The Kaplan–Meier method was used to assess RFS and CSS among patients, stratified by the different genomic events. Multivariate Cox proportional hazards model was also used to test the differences in survival outcomes across the studied genomic features with the adjustment of age at diagnosis, sex, presurgery and post-surgery RT. All statistical tests in the present study were two sided and performed using SAS version 9.4 (SAS Institute, Cary, NC, USA) or R version 3.6.1 (R Foundation for Statistical Computing, Vienna, Austria).

**Reporting summary**. Further information on research design is available in the Nature Research Reporting Summary linked to this article.

## Data availability
The WGS and RNA-Seq data generated in this study have been deposited in the dbGaP database under Accession Code phs002301.v1.p1. The remaining data are available within the article, Supplementary information, or available from the authors upon request.

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

## Acknowledgements

This research was supported by Beijing Municipal Science and Technology Commission (Z171100000117002), Research Special Fund for Public Welfare Industry of Health (201402008), and the Intramural Research Program of the National Institutes of Health, National Cancer Institute, Division of Cancer Epidemiology and Genetics.

## Author contributions

All authors of this research paper have directly participated in the planning, execution, or analysis of the study. Specifically, J.B., J.S., C.L., X.R.Y., and Y. Zhang contributed to the study conception and design. J.B., C.L., S.W., Y. Zhai, M.L., S.Z., Q.L., P.Z., S.G., and Y. Zhang contributed to patient recruitment, data and specimen collection, tissue processing, and experimental assays. J.D. and J.W. helped with the pathologic review. T.Z., B.Z.(bioinformatician), L.S., D.W., M.W., W.Z., and B.J.B. performed bioinformatics analyses. J.S., X.H., H.K., H.H.W., and B.Z. (biostatistician) conducted all the statistical analyses. B.H. and Y. Zhang provided administrative support. L.M. helped with germline rare variant classification. D.M.P. and A.M.G. provided expertise in genetic epidemiology of chordoma. J.B., J.S., and X.R.Y. were responsible for data interpretation and drafted the paper. Y. Zhang secured the funding for the sequencing analysis. X.R.Y. provided supervision in all components of the study from study design to data interpretation and critical revision of the paper.

## Competing interests

The authors declare no competing interests.
