## [Peer Review File · Nature Communications]

REVIEWER COMMENTS

Reviewer #1 (Remarks to the Author): Expert in chordoma

The authors have performed whole genome sequencing (WGS) on a series of 80 skull base chordomas (SBCs) of differing histology. They identified genomic alterations in SWI/SNF, PBRM1, SETD2, and the CDKN2A/2B locus. As stated by the authors this represents the largest WGS analysis of SBCs to date. The manuscript is well written, and the data analysis straightforward.

There have been several previous molecular genetic analyses of chordomas, many of which the authors have cited, but several have not:

Neoplasia. 2012 Sep;14(9):788-98. CDKN2A/2B locus known a while ago

J Pediatr Hematol Oncol. 2020 Apr;42(3):218-219. doi: 10.1097/MPH.0000000000001721.

Pediatr Dev Pathol. 2018 Jan-Feb;21(1):6-28. doi: 10.1177/1093526617749671.

These and other studies have indicated the role of the CDKN2A/2B locus, the T gene duplication, and the SWI/SNF complex.

The authors have attempted to correlate a relationship between gene alterations and overall chordoma behaviour/prognosis. Unfortunately, without knowing the protocols for treatment of patients in this study, in particular the type of surgery performed, the type of radiation therapy delivered, and other adjuvant therapies that may have been employed, it is very difficult to make such a correlation.

There is a possibility for patient selection bias as we do not know how patients were determined for inclusion in their study.

The analysis of tumor mutation burden and patients who received pre-surgery radiation therapy is subject to criticism of bias.

In the end candidate driver genes were identified in only 27 of 80 tumors. In the other tumors, the mechanism of tumor pathogenesis is unknown and subject to some speculation.

What is needed in a study such as this is prospective validation, or external validation with a cohort of patients from another institution.

While the authors should be commended on performing WGS on such a large number of patients with SBCs, it is not clear that they have reliably identified molecular markers that correlate with clinical outcomes or patient prognosis for the reasons mentioned above.

Reviewer #2 (Remarks to the Author):Expert in chordoma

Congratulations to the researchers and clinical team of compiling such a large collection of this rare tumour.

This is a well written manuscript describing comprehensively the genomic landscape of skull base chordomas.

The data provided will be a valuable resource for the research community.

I offer a number of comments

1. To ensure that all of your cases represent chordoma it would be useful to confirm the diagnosis by showing brachyury / TBXT (the term T is no longer recommended) expression on immunohistochemistry. I apologise if I missed that you have done this. Specifically it is important to distinguish from cartilaginous tumours. I note that one of the chordomas harboured a IDH1 SNV - a genetic alteration which is characteristic of a central cartilaginous tumour. I am sure that this will be confirmed when you undertake immuno studies and that you will see the lack of expression TBXT and cytokeratin - hallmarks of Chordoma - other than the dedifferentiated variant. Only about 50% of central cartilaginous tumour have an IDH1/2 mutation so failure to detect this genetic alteration could result in missing a diagnosis of a cartilaginous tumour. Hence it would be useful to review off the immunostains to ensure that brachyury is expressed.

2. Following on from the comment above you state in the Discussion - page 20, line 449/50 that the expression of TBXT in chordoma is a poor prognosis. Expression of brachyury on immunostaining is a requirement to make a diagnosis of chordoma - others than dediff chordomas - see the WHO Blue book on bone tumours. Any pathologist today would confirm this. Either the cases in these published studies that are cited in the paper were not chordomas or more likely that the decalcification process resulted in tissue damage and the absence of immunoreactivity. I would prefer if this short paragraph were removed from the MS as I believe will result in persistent wrong information about how a diagnosis of chordoma is made and lead to misdiagnosis of patients' tumours.

3. It is great to see that all of the samples have come from one centre and received the same standard of care. However, can you please clarify if all of the 80 patients had their first operation in the Neurosurgery Department of Beijing Tiantan Hospital, Capital Medical University; that is, were any of these patients treated in another centre before being having surgery at Beijing Tiantan Hospital? This is important as it would impact on the survival. It is not infrequent that chordomas are misdiagnosed (particularly as the cases go back as far as 2010 and before the use of immuno was introduced as a standard of care in making this diagnosis) and treated in non-specialist hospitals and as a general the outcome for such patients is poor.

4. I note that very few patients were reported to have metastatic disease - how earnestly were the patients screened? The literature and my personalised experience is that most patients develop metastatic disease but because there is no treatment of systemic disease, metastatic lesions are not sought and therefore if some develop they may not be reported at the patient may well be in a palliative case unit or at home remote from where the surgery was undertaken

5. I liked the way that the Somatic copy number alterations (SCNAs) were divided into groups and was interested to see that 19 of the cases (25%) had essentially no SCNA; Did these cases have significant SNVs? It found it difficult to interrogate the SCNAs and SNVs case by case so may be this could be made more easy for the reader? Could it be possible that some / a few of these 'chordomas' represent benign notochordal tumours or even echordosis physaliphora?

6. Detection of copy number gain of TBXT in only 2% is very small compared to other reports, as acknowledged by the authors. I suspect that this is likely to be explained on the basis that as shown by others, that gain is generally very minor (one or 2 extra copied). The authors cite the paper by Tarpey et al, but in this study the findings were only detected in the extended study when targeted high depth sequencing was performed. The finding of TBXT CNG in ~20% has also been demonstrated by FISH where it is feasible to detect just one or two extra copies. I suggest that it would be important to convey this in the paper.

7. It would be interesting to know if this cohort carried the rs2305089 SNP in TBXT, which is present in 97% of European patients with chordoma.

6. Discussion I feel is too long. I would like to see it reduce by a good 50%.

Reviewer #3 (Remarks to the Author): Expert in genomics

The authors provide a detailed characterisation of the genomic landscape of chordoma affecting the skull-base, a rare but interesting cancer that has only been patchily studied at the genomic level. The paper comprehensively addresses this, with an impressive sample size given the rarity of the condition. There is much to like in this paper, and the analyses are well-performed using the community-accepted standards. I think it will be of interest to the readership of Nature Communications.

I have the following relatively minor comments -

1. One of the things that has always puzzled me about chordoma is the relatively high fraction of patients with no identified driver mutations. This study confirms this. I think it would be worth drawing this point out in more detail in the manuscript, and providing some commentary / analysis on the group of patients with no identified drivers. Are they more likely to have consistent and severely abnormal patterns of chromosomal aneuploidy (as we recently found for chromophobe kidney cancers and PNET tumours in PCAWG)? Do they have high or low tumour mutational burden?

2. In the survival analysis, is it possible to propose a multi-variable prognostic model that allows accurate discrimination of high or low-risk patients?

Signed, Peter Campbell

Reviewer #4 (Remarks to the Author):Expert in genomics

The paper by Bai et al., comprehensively characterized the genomic landscape of 80 skull-base chordomas and identified recurrent SWI/SNF gene driver mutations in PBRM1. Chordoma has low tumor mutational burden, with about 50% of samples with unknown genomic drivers. Mutational landscape is heterogenous. The authors conducted one of the largest cohorts of chordoma by WGS and found significant somatic mutation, structural variant and rearrangement of PBRM1 and SETD2. Interestingly, SWI/SNF alterations (? SETD2 included) are significantly associated with worse chordoma-specific and recurrence-free survival. Overall, the manuscript presented high quality data of WGS of one of the largest cohorts of skull-based chordomas. However, it is mainly descriptive and lacks the novelty and mechanistic insights on how PBRM1/SETD2 contribute to the tumorigenesis/prognosis of chordoma.

Major concerns:

1. SETD2 is a chromatin modifier, but not traditionally considered a SWI/SNF complex genes. In the manuscript, SETD2 is described together with PBRM1 as SWI/SNF complex genes.
2. The authors compared the CSS and RFS between chordomas with and without SWI/SNF gene alterations; after adjusting for age, sex, pre/post-surgery RT, they found that the the SWI/SNF altered chordoma had worse outcome. In this analysis, they included SETD2 which is not part of the SWI/SNF complex. Importantly, what are the potential mechanisms for the worse prognosis? Do SETD2 and the SWI/SNF complexes gene (PBRM1 and ARID2, etc) have independent prognostic role? The SWI/SNF/SETD2 subsets are associated with more SNVs and CNVs. Would these influence the prognosis rather than PBRM1 and/or SETD2 per se?

REVIEWER COMMENTS

Reviewer #1 (Remarks to the Author): Expert in chordoma

The authors have performed whole genome sequencing (WGS) on a series of 80 skull base chordomas (SBCs) of differing histology. They identified genomic alterations in SWI/SNF, PBRM1, SETD2, and the CDKN2A/2B locus. As stated by the authors this represents the largest WGS analysis of SBCs to date. The manuscript is well written, and the data analysis straightforward.

Response: Thank you.

1. There have been several previous molecular genetic analyses of chordomas, many of which the authors have cited, but several have not:

Neoplasia. 2012 Sep;14(9):788-98. CDKN2A/2B locus known a while ago

J Pediatr Hematol Oncol. 2020 Apr;42(3):218-219. doi: 10.1097/MPH.0000000000001721.

Pediatr Dev Pathol. 2018 Jan-Feb;21(1):6-28. doi: 10.1177/1093526617749671.

These and other studies have indicated the role of the CDKN2A/2B locus, the T gene duplication, and the SWI/SNF complex.

Response: Thank you for sharing these references! We have now cited them in Results and Discussion in the revised manuscript.

2. The authors have attempted to correlate a relationship between gene alterations and overall chordoma behaviour/prognosis. Unfortunately, without knowing the protocols for treatment of patients in this study, in particular the type of surgery performed, the type of radiation therapy delivered, and other adjuvant therapies that may have been employed, it is very difficult to make such a correlation.

Response: The vast majority of our patients had endoscopic endonasal surgery, with the exception of only two patients who had open craniotomy. None of these patients received any chemo, immune, or targeted therapies. As we included in Table 1, 12 of these patients had pre-surgery radiation therapy (RT) and 42 had post-surgery RT, and we adjusted for both pre-surgery and post-surgery RT in our survival models. Since RT was performed at multiple institutes/clinics, the detailed data on RT type, duration and dosage is not available to us. We have added this as a limitation in the Discussion. To further account for potential confounders, we have also included the gross resection rate and Ki-67 in our survival analyses but results did not change noticeably. We also conducted a sensitivity analysis by restricting to the 53 patients who were not previously diagnosed with or treated for chordoma and who did not have pre-surgery RT and we found that the associations for PBRM1 alterations and 22q deletion for both CSS and RFS remain significant (they became even stronger). We have included the additional clinical information in Table 1 and added Suppl Table 6c showing survival results from sensitivity analyses.

3. There is a possibility for patient selection bias as we do not know how patients were determined for inclusion in their study.

Response: Since chordoma is a rare cancer, we try to collect tumor tissue samples from all patients who were operated by us. We conducted immunohistochemical staining on tumor samples we collected to confirm chordoma diagnosis and to check tumor cell content. We selected samples for WGS based on tumor content and DNA quality, not based on patient clinical information. Therefore, the selection bias should be minimal.

4. The analysis of tumor mutation burden and patients who received pre-surgery radiation therapy is subject to criticism of bias.

Response: We agree with the reviewer on this point. As we described in the manuscript (Results under Somatic Genomic Landscape), we did see that patients who received pre-surgery RT (N=12) tended to have higher TMB (median=0.75 mutations/Mb, range=0.055 to 1.56) compared with patients without pre-surgery treatment (N=68, median=0.49 mutations/Mb, range=0.05 to 7.68), however, the difference was not statistically significant (p=0.42). To address the reviewer's concern, we conducted sensitivity analyses by removing the 12 patients with pre-surgery RT from all TMB related analyses and the results did not change noticeably. For example, associations between mutation burden and alterations in PBRM1 (p=0.12) and CDKN2A/2B (p=0.06) were similar but less significant (probably due to smaller sample size) when restricting to patients without pre-surgery RT. We included these results in the revised manuscript.

5. In the end candidate driver genes were identified in only 27 of 80 tumors. In the other tumors, the mechanism of tumor pathogenesis is unknown and subject to some speculation.

Response: As we mentioned in the summary of driver events in the Results section, we think that "The remaining tumors might be caused by non-synonymous mutations or SCNAs/SVs in known cancer driver genes (observed in 8% patients, Figure 2) or other driver genes/mechanisms for which statistical power was too low to detect in this study." These other mechanisms may include SCNAs/SVs or epigenetic events. In particular, some of these tumors showed extensive chromosomal SCNAs that are consistent with chromosomal aneuploidy (shown in newly added Supplementary Figure 7), which may drive chordoma development, similar to what has been previously reported in chromophobe renal cell carcinomas and pancreatic neuroendocrine tumors that showed high fractions of tumors without known drivers (ICGC & TCGA, Nature, 2020). In addition, we identified several significant focal SCNA regions, which may harbor driver events. However, with the exception of 9p21.3 containing CDKN2A/2B and 3p21.1 containing PBRM1 and SETD2, most other significant focal SCNA regions did not contain known cancer driver genes, and therefore, it is challenging to identify the driver genes. Previous Pan-cancer analysis of SCNAs found that significant focal SCNAs without known cancer genes were enriched with genes involved in epigenetic regulation (Zack et al., Nat Genet, 2013). Future mechanistic studies should follow up genes in these focal SCNA regions to identify potential driver genes. Further, multiple groups are actively investigating the epigenomic landscape of chordoma and results from these studies will

hopefully shed lights on the role of epigenetic mechanisms in driving chordoma development, which has long been suspected in the field. We have included this in the Results and Discussion in the revised manuscript.

6. What is needed in a study such as this is prospective validation, or external validation with a cohort of patients from another institution.

While the authors should be commended on performing WGS on such a large number of patients with SBCs, it is not clear that they have reliably identified molecular markers that correlate with clinical outcomes or patient prognosis for the reasons mentioned above.

Response: Although we agree with the reviewer that prognostic markers identified in this study should be validated by other studies before they can be implemented in clinic settings, it is extremely challenging to replicate the results using whole-genome sequencing analysis especially in a prospective study since it would require the recruitment of additional patients which will take years for such a rare cancer. To determine whether data to evaluate this question is available publicly, we contacted the Chordoma Foundation and Foundation Medicine. Unfortunately, Chordoma Foundation has very little clinical outcome data for patients with tissue samples available. Similarly, Foundation Medicine sequenced a relatively large number of chordoma tumors (>200), but treatment and outcome data are not available. To this end, we think it'd be very helpful for us to share the findings with the research community so that these candidate markers can be followed up and externally validated.

Reviewer #2 (Remarks to the Author):Expert in chordoma

Congratulations to the researchers and clinical team of compiling such a large collections of this rare tumour. This is a well written manuscript describing comprehensively the genomic landscape of skull base chordomas. The data provided will be a valuable resource for the research community.

Response: Thank you.

I offer a number of comments

1. To ensure that all of your cases represent chordoma it would be useful to confirm the diagnosis by showing brachyury / TBXT (the term T is no longer recommended) expression on immunohistochemistry. I apologised if I missed that you have done this. Specifically it is important to distinguish from cartilaginous tumours. I note that one of the chordomas harboured a IDH1 SNV - a genetic alteration which is characteristic of a central cartilaginous tumour. I am sure that this will be confirmed when you undertake immuno studies and that you will see the lack of expression TBXT and cytokeratin - hallmarks of Chordoma - other than the dedifferentiated variant. Only about 50% of central cartilaginous tumour have an IDH1/2 mutation so failure to detect this genetic alteration could result in missing a diagnosis of a cartilaginous tumour. Hence it would be useful to review off the immunostains to ensure that brachyury is expressed.

Response: Brachyury staining has become a routine practice in diagnosing chordoma diagnosis in more recent years. Before using Brachyury as a diagnostic marker, we routinely stained tumor samples with

cytokeratins and EMA to help us distinguish chordomas from cartilaginous tumors or chondrosarcomas. Following the reviewer's suggestion, we carefully checked immunostaining results and stained all patients who did not have Brachyury staining data and had tissue sections available for Brachyury. After this effort, we managed to obtain Brachyury staining data for 70 patients, and we confirmed the chordoma diagnosis in these patients. We used morphology in combination with CK and EMA markers to make the chordoma diagnosis for the remaining 10 patients. Of the two tumors with IDH1 mutations, Brachyury staining data is available for one of them, which confirmed the chordoma diagnosis. The diagnosis of chordoma in another tumor for which Brachyury data was not available was confirmed by the careful re-review of morphology and other relevant data by our pathologist. We included this information in the Methods section of the revised manuscript.

We have changed T to TBXT throughout the manuscript.

2. Following on from the comment above you state in the Discussion - page 20, line 449/50 that the expression of TBXT in chordoma is a poor prognosis. Expression of brachyury on immunostaining is a requirement to make a diagnosis of chordoma - others than dediff chordomas - see the WHO Blue book on bone tumours. Any pathologist today would confirm this. Either the cases in these published studies that are cited in the paper were not chordomas or more likely that the decalcification process resulted in tissue damage and the absence of immunoreactivity. I would prefer if this short paragraph were removed from the MS as I believe will result in persistent wrong information about how a diagnosis of chordoma is made and lead to misdiagnosis of patients' tumours.

Response: Please see our response to the question related to the diagnosis and TBXT staining. We agree with the reviewer that the sentence describing the findings from the referenced study is confusing and we removed that paragraph accordingly.

3. It is great to see that all of the samples have come from one centre and received the same standard of care. However, can you please clarify if all of the 80 patients had their first operation in the Neurosurgery Department of Beijing Tiantan Hospital, Capital Medical University; that is, were any of these patients treated in another centre before being having surgery at Beijing Tiantan Hospital? This is important as it would impact on the survival. It is not infrequent that chordomas are misdiagnosed (particularly as the cases go back as far as 2010 and before the use of immuno was introduced as a standard of care in making this diagnosis) and treated in non-specialist hospitals and as a general the outcome for such patients is poor.

Response: The reviewer raised a very important question. Indeed, this patient cohort included some patients from other hospitals and were treated before. In addition, some patients who were diagnosed in our hospital were also treated with radiation therapy before surgeries. We therefore adjusted for pre-surgery RT in our survival models. To further assess whether these factors might have influenced the associations with disease outcomes, we performed sensitivity analyses restricting to patients who were diagnosed and operated in our hospital and did not have any treatment prior to surgeries. We found that the associations for PBRM1 alterations and 22q deletion for both CSS and RFS remain significant (they became even stronger), and we have included these results in newly added Supplementary Table 6c.

4. I note that very few patients were reported to have metastatic disease - how earnestly were the patients screened? The literature and my personalised experience is that most patients develop metastatic disease but because there is no treatment of systemic disease, metastatic lesions are not sought and therefore if some develop they may not be reported at the patient may well be in a palliative case unit or at home remote from where the surgery was undertaken

Response: The rate of metastasis among skull-base chordomas is lower compared to sacral chordomas, as previously reported (Di Maio S, et al., JNS, 2011). Metastases in our patients are usually identified either when patients present with symptoms or through ultrasound and CT scan in distant organs. It is possible that metastasis was missed in some patients. However, given that metastasis among skull-base chordoma patients usually occurs at advanced stages and the follow up time in our WGS cohort was not very long, we believe that most patients had not developed metastases. In our retrospective cohort in which we have followed up 284 skull-base chordoma patients for an average of 44 months, nine patients (3.2%) developed metastases, which is still relatively low. Based on our experience, metastases among adult patients usually occur >6 years after surgery, while children may develop metastases in a shorter period of time. What remains unclear is whether there are racial differences in metastasis rates of skull-base chordoma.

5. I liked the way that the Somatic copy number alterations (SCNAs) were divided into groups and was interested to see that 19 of the cases (25%) had essentially no SCNA; Did these cases have significant SNVs? It found it difficult to interrogate the SCNAs and SNVs case by case so may be this could be made more easy for the reader? Could it be possible that some / a few of these 'chordomas' represent benign notochordal tumours or even echordosis physaliphora?

Response: The patients with few SCNA events also had lower number of SNVs as compared to patients with extensive SCNA events. However, about a third of these patients harbored a potential driver event, for example, four patients had focal 9p21 homologous deletion or SV affecting CDKN2A and 1 patient had PBRM1 mutation. We have now added the SCNA group in Figure 2 to make it easier to capture the relationship between SNVs and SCNAs. We have carefully examined the clinical characteristics of these patients and confirmed the chordoma diagnosis in these patients. In particular, benign notochordal tumours or echordosis physaliphoras have distinct cell morphology such as lower extracellular matrix content, adipocyte-like features, and lower amount of tumor nuclei as compared to chordomas. Although benign tumors also stain positive for brachyury, they usually have very low Ki67 levels. In addition, we also examined MR imaging signal intensity and electron ultramicroscopic features to confirm the chordoma diagnosis in our patients (Bai J, et al., World Neurosurg, 2017; Bai J, et al., Am J Neuroradio, 2020).

6. Detection of copy number gain of TBXT in only 2% is very small compared to other reports, as acknowledged by the authors. I suspect that this is likely to be explained on the basis that as shown by others, that gain is generally very minor (one or 2 extra copied). The authors cite the paper by Tarpey et al, but in this study the findings were only detected in the extended study when targeted high depth sequencing was performed. The finding of TBXT CNG in ~20% has also been demonstrated by FISH

where it is feasible to detect just one or two extra copies. I suggest that it would be important to convey this in the paper.

Response: We agree with the reviewer that assay technologies may influence the sensitivity of SCNA detection for low-copy gains. In addition, variations in how SCNAs are defined (arm/chromosome-level vs. focal) are also likely to cause various frequencies of TBXT copy gains reported in the literature. In our study, although one-copy 6q gain was observed in 9 samples (11.25%), focal TBXT amp was only seen in two patients. A genome-wide SNP genotyping array analysis of skull-base chordoma reported a similarly low frequency of focal TBXT amp (1 of 18 samples) (Diaz, 2012). In contrast, in Tarpey's study, focal T amp was reported in 3 out of 11 WGS samples, suggesting that focal TBXT amplifications may be more common in sacral than skull-base chordomas. Nevertheless, high-resolution assays of this region in large studies including chordomas of different sites are needed to solve these discrepancies. We have added this to the discussion in the revised manuscript.

7. It would be interesting to know if this cohort carried the rs2305089 SNP in TBXT, which is present in 97% of European patients with chordoma.

Response: Following the reviewer's suggestion, we extracted this SNP from the germline sequencing data. The allele frequency of the variant allele (A) is 33.7% in this patient cohort, which is very similar to what was reported in 1k genome and ExAC among East Asians. Unlike in the European populations, where the variant allele frequency (AF) is ~50% in the general population and >90% in chordoma cases, AF is much lower in the East Asian population and it does not seem to vary in chordoma cases and unaffected people, which is consistent with the findings from a previous Chinese study of skull-base chordoma (Wu et al., Int. J. Mol. Sci. 2013). We further checked whether this SNP is related to any somatic genomic features or clinical outcomes in our patient cohort, and we did not observe any significant associations. We included these results in the revised manuscript.

6. Discussion I feel is too long. I would like to see it reduce by a good 50%.

Response: Following the reviewer's suggestion, we have shortened the discussion significantly, please see track changes for our effort. However, we think it is also helpful to provide our view on potential implications of the results, especially those that are related to etiology (mutational signatures), treatment (HRD and PARP1, PBRM1), and prognosis (intriguing results of 9q vs. 9p and 22q), as well as comparisons to results based on sacral chordoma, and we therefore kept the discussions on these points. In addition, we also added some topics in the discussion to respond to other reviewers' comments.

Reviewer #3 (Remarks to the Author):Expert in genomics

The authors provide a detailed characterisation of the genomic landscape of chordoma affecting the skull-base, a rare but interesting cancer that has only been patchily studied at the genomic level. The paper comprehensively addresses this, with an impressive sample size given the rarity of the condition. There is much to like in this paper, and the analyses are well-performed using the community-accepted standards. I think it will be of interest to the readership of Nature Communications.

Response: Thank you.

I have the following relatively minor comments -

1. One of the things that has always puzzled me about chordoma is the relatively high fraction of patients with no identified driver mutations. This study confirms this. I think it would be worth drawing this point out in more detail in the manuscript, and providing some commentary / analysis on the group of patients with no identified drivers. Are they more likely to have consistent and severely abnormal patterns of chromosomal aneuploidy (as we recently found for chromophobe kidney cancers and PNET tumours in PCAWG)? Do they have high or low tumour mutational burden?

Response: Thank you so much for providing this insight, which is very helpful! Indeed, the extensive chromosomal SCNAs were observed in tumors without known drivers, including five of seven patients with whole genome doubling. We agree with the reviewer that these events may be sufficient as driver events in tumors without focal mutations. We added a supplementary figure (new Supplementary Figure 7) showing chromosomal aneuploidy in tumors without known drivers to illustrate this point and included some discussions. Please also see our response to Question 5 by Reviewer 1 on this topic. In summary, we think there are multiple possibilities for those patients without driver events identified in our study. First, some remaining samples are due to rare mutations, however, our sample size is limited to identify rare driver mutations if the driver gene landscape is very heterogeneous in chordoma. As we mentioned in the summary of driver events in the Results section, we think that “The remaining tumors might be caused by non-synonymous mutations or SCNAs/SVs in known cancer driver genes (observed in 8% patients, Figure 2) or other driver genes/mechanisms for which statistical power was too low to detect in this study.” These other mechanisms may include SCNAs/SVs or epigenetic events. In particular, some of these tumors showed extensive chromosomal SCNAs that are consistent with chromosomal aneuploidy (shown in newly added Supplementary Figure 7), which may drive chordoma development, similar to what has been previously reported in chromophobe renal cell carcinomas and pancreatic neuroendocrine tumors that showed high fractions of tumors without known drivers (ICGC & TCGA, Nature, 2020). In addition, we identified several significant focal SCNA regions, which may harbor driver events. However, with the exception of 9p21.3 containing CDKN2A/2B and 3p21.1 containing PBRM1 and SETD2, most other significant focal SCNA regions did not contain known cancer driver genes, and therefore, it is challenging to identify the driver genes. Previous Pan-cancer analysis of SCNAs found that significant focal SCNAs without known cancer genes were enriched with genes involved in epigenetic regulation (Zack et al., Nat Genet, 2013). Future mechanistic studies should follow up genes in these focal SCNA regions to identify potential driver genes. Further, multiple groups are actively investigating the epigenomic landscape of chordoma and results from these studies will hopefully shed lights on the role of epigenetic mechanisms in driving chordoma development, which has long been suspected in the field. We have included this in the Results and Discussion in the revised manuscript.

2. In the survival analysis, is it possible to propose a multi-variable prognostic model that allows accurate discrimination of high or low-risk patients?

Response: We appreciate the suggestion of developing a multi-variable prognostic model and we consider this as an ultimate goal of this type of analyses. However, since we have tested multiple (~20) genomic features to identify PBRM1 and deletions of 9p21.3, 9q21.11 and 22q as significant risk factors

for CSS/PFS, refitting a multivariable model including only these variables would lead to biased estimates of HR. Moreover, the relatively small sample size (due to the rarity of the disease) makes it a challenge to split the data into discovery (identifying significant risk factors) and validation (fitting the model) datasets. A formal, unbiased multivariable model would be obtained by fitting the model in an independent dataset. As for now, a patient with alterations in these genes/regions can be considered as having a higher risk of a worse outcome, although we are not able to provide an unbiased estimate of the risk because of the winner's curse.

Reviewer #4 (Remarks to the Author):Expert in genomics

The paper by Bai et al., comprehensively characterized the genomic landscape of 80 skull-base chordomas and identified recurrent SWI/SNF gene driver mutations in PBRM1. Chordoma has low tumor mutational burden, with about 50% of samples with unknown genomic drivers. Mutational landscape is heterogeneous. The authors conducted one of the largest cohorts of chordoma by WGS and found significant somatic mutation, structural variant and rearrangement of PBRM1 and SETD2. Interestingly, SWI/SNF alterations (? SETD2 included) are significantly associated with worse chordoma-specific and recurrence-free survival. Overall, the manuscript presented high quality data of WGS of one of the largest cohorts of skull-based chordomas. However, it is mainly descriptive and lacks the novelty and mechanistic insights on how PBRM1/SETD2 contribute to the tumorigenesis/prognosis of chordoma.

Response: We thank the reviewer for the comments. We agree with the reviewer that this is a descriptive study but we think it is important to share the findings with the research community so that the genes/regions of interest can be followed up by those investigators who have the right approach/assays (such as genetic mouse model) to further address these questions.

Major concerns:

1. SETD2 is a chromatin modifier, but not traditionally considered a SWI/SNF complex genes. In the manuscript, SETD2 is described together with PBRM1 as SWI/SNF complex genes.

Response: Thank you for pointing this out. We removed the three SETD2+ patients from the SWI/SNF+ group and updated all analyses, texts, and figures to reflect the change.

2. The authors compared the CSS and RFS between chordomas with and without SWI/SNF gene alterations; after adjusting for age, sex, pre/post-surgery RT, they found that the the SWI/SNF altered chordoma had worse outcome. In this analysis, they included SETD2 which is not part of the SWI/SNF complex. Importantly, what are the potential mechanisms for the worse prognosis? Do SETD2 and the SWI/SNF complexes gene (PBRM1 and ARID2, etc) have independent prognostic role? The SWI/SNF/SETD2 subsets are associated with more SNVs and CNVs. Would these influence the prognosis rather than PBRM1 and/or SETD2 per se?

Response: We have taken out SETD2+ patients from SWI/SNF+. To address whether the identified significant features (PBRM1 and deletions of 9p21.3, 9q21.11 and 22q) were associated with CSS/RFS

independently of TMB or SCNAs, we added TMB and SCNA group in survival models and results were very similar after accounting for these variables. The estimate for 22q deletion associated with CSS became stronger. We have added a sentence describing this result in the revised manuscript and added Suppl Table 6c to present these results.

REVIEWERS' COMMENTS

Reviewer #1 (Remarks to the Author):

The authors have satisfactorily answered my queries.

Reviewer #2 (Remarks to the Author):

See my comment to authors -

Reviewer #3 (Remarks to the Author):

The authors have responded comprehensively to my comments and I have no further suggestions for improvement.

Reviewer #4 (Remarks to the Author):

This is a significantly improved manuscript. However, I think the findings remain descriptive.

REVIEWER COMMENTS

Reviewer #2 continued to raise concerns with regards to the IDH1 mutant status of two of your tumours, therefore we asked Reviewer #1 with a similar expertise to comment on this issue. I am afraid that this also slightly added to the delay. Reviewer #1 agreed that it is important to consider that these tumours might be chondrosarcomas and encouraged you to discuss this point.

Response: As we explained in our previous response to the reviewer's question, of the two tumors with IDH1 mutations, Brachyury staining data was available for one of them (Supplementary figure 10 b), and the positive staining confirmed the chordoma diagnosis. The diagnosis of chordoma in another tumor for which Brachyury data was not available was confirmed by the careful re-review of morphology and positive staining for cytokeratins and EMA (Supplementary figure 10 d-f), which are epithelial cell markers to help us distinguish chordomas from cartilaginous tumors or chondrosarcomas. We have included a figure to show the H&E and immunohistochemical staining images of these two tumors. We would be happy to include it as a supplementary figure if the editor and reviewer think it is necessary.

Figure legend

Images of haematoxylin and eosin (H&E) and immunohistochemical (IHC) staining of the two patients with IDH1 mutations (**a-c** patient P59, **d-f** patient P26). **a** and **d**: H&E staining; **b**: High expression of BRACHYURY; **c** and **f**: Strong positive for CYTOKERATIN; **e**: Positive EMA staining.